

# First results of the Piton de la Fournaise STRAP 2015 experiment: multidisciplinary tracking of a volcanic gas and aerosol plume

Pierre Tulet[1], Andréa Di Muro[2], Aurélie Colomb[3], Cyrielle Denjean[4], Valentin Duflot[1], Santiago Arellano[5], Brice Foucart[1,3], Jérome Brioude[1], Karine Sellegri[3], Aline Peltier[2], Alessandro Aiuppa[6,7], Christelle Barthe[1], Chatrapatty Bhugwant[8], Soline Bielli[1], Patrice Boissier[2], Guillaume Boudoire[2], Thierry Bourrianne[4], Christophe Brunet[2], Fréderic Burnet[4], Jean-Pierre Cammas[1,9], Franck Gabarrot[9], Bo Galle[5], Gaetano Giudice[7], Christian Guadagno[8], Fréderic Jeamblu[1], Philippe Kowalski[2], Jimmy Leclair de Bellevue[1], Nicolas Marquestaut[9], Dominique Mékies[1], Jean-Marc Metzger[9], Joris Pianezze[1], Thierry Portafaix[1], Jean Sciare[10], Arnaud Tournigand[8], and Nicolas Villeneuve[2]

[1]LACy, Laboratoire de l'Atmosphère et des Cyclones (UMR8105 CNRS, Université de La Réunion, Météo-France), Saint-Denis de La Réunion, France
[2]OVPF, Institut de Physique du Globe de Paris (UMR7154, CNRS, Université Sorbonne Paris-Cité, Université Paris Diderot), Bourg-Murat, La Réunion, France
[3]LaMP, Laboratoire de Météorologie Physique (UMR6016, CNRS, Université Blaise Pascal), Clermont-Ferrand, France
[4]CNRM, Centre National de la Recherche Météorologique (UMR3589, CNRS, Météo-France), Toulouse, France
[5]DESS, Department of Earth and Space Sciences, Chalmers University of Technology, Gothenburg, Sweden
[6]Dipartimento DiSTeM, Universitá di Palermo, Italy
[7]INGV, Istituto Nazionale di Geofisica e Vulcanologia, Sezione di Palermo, Italy
[8]ORA, Observatoire Réunionais de l'Air, Saint-Denis de La Réunion, France
[9]OSU-R, Observatoire des Sciences de l'Univers de la Réunion (UMS3365 CNRS, Université de La Réunion), Saint-Denis de La Réunion, France
[10]Energy, Environment, Water, Research Center, The Cyprus Institute, Nicosia, Cyprus

*Correspondence to:* P. Tulet
(pierre.tulet@univ-reunion.fr)

**Abstract.** The STRAP (Synergie Transdisciplinaire pour Répondre aux Aléas liés aux Panaches volcaniques) campaign was conducted in 2015 to investigate the volcanic plumes of Piton de La Fournaise (La Réunion, France). For the first time, measurements at the local (near the vent) and at the regional scales around the island were conducted. The STRAP 2015 campaign has become possible thanks to a strong cross-disciplinary collaboration between volcanologists and meteorologists. The main observations during four eruptive periods (85 days) are summarized. They include the estimates of $SO_2$, $CO_2$ and $H_2O$ emissions, the altitude of the plume at the vent and over different areas of La Réunion Island, the evolution of the $SO_2$ concentration, the aerosol size distribution, and the aerosol extinction profile. A climatology of the volcanic plume dispersion is also reported. Simulations and measurements showed that the plume formed by weak eruption has a stronger interaction with the surface of the island. Strong $SO_2$ and particles concentrations above 1000 $ppb$ and 50,000 $cm^{-3}$, respectively, are frequently measured over 20 $km$ of distance from the Piton de la Fournaise. The measured aerosol size distribution shows the predominance of small particles in the volcanic plume. A particular emphasis is placed on the gas-particle conversion with several cases of strong nucleation of sulfuric acid observed within the plume and at the distal site of the Maïdo observatory.



The STRAP 2015 campaign gave a unique set of multi-disciplinary data that can now be used by modellers to improve the numerical paramameterizations of the physical and chemical evolution of the volcanic plumes.

## 1 Introduction

The 2010 eruption of Eyjafjallajökull (Iceland) showed how a relatively small volcanic event can directly impact the life of
millions of people. Volcanic plumes can cause environmental, economic and societal impacts, and better knowledge on their physics, chemistry and time evolution are of paramount importance to improve the quality of our predictions and forecasts. On one side, improving our ability to quantify and model the genesis, dispersion and impact of a volcanic plume is thus a key challenge for scientists and societal stakeholders. On the other side, mitigation of volcanic crisis relies on efficient, and effective, communication and interaction between multidisciplinary scientific actors in geology, physics, chemistry, and remote
sensing. The ultimate goal of the multidisciplinary approach is to couple a precise and real-time quantification of the volcanic source terms with accurate and fast modelling of physical and chemical processes to predict the ascent, dispersion and impact of volcanic ash and gas in the atmosphere. Eyjafjallajökull demonstrated in an emblematic way the challenges of forecasting volcanic source parameters, the temporal and spatial evolution of a relatively weak volcanic plume, as well as of accurately assessing the ash and gas content, composition and size distribution of the drifting mass (Kaminski et al., 2011; Bursik et al.,
2012; Folch et al., 2012). One major source of uncertainty is the lack of accurate measurements of basic and essential parameters, such as plume height and mass discharge rate. In spite of the critical importance of source parameters, Eyjafjallajökull showed that in practice it is difficult and sometimes impossible to obtain and communicate with the needed accuracy, especially in the time frame of an eruptive crisis (i.e., hours to days). Indeed, for Volcanic Ash Advisory Centers (VAACs), which are the organizations in charge of ash cloud dispersion forecasts, such measurements have clearly been identified as critical if forecast
models are to be run efficiently (Bonadonna et al., 2012). The models themselves also still need major improvements if we are to adequately take into account the complexity, and rapidly changing dynamics of a volcanic cloud. If an improvement to our understanding of processes controlling the formation and evolution of volcanic clouds can be achieved it will represent a major step forward for the scientific community, as well as for socio-political stakeholders.

Volcanic emissions are an important factor for climate change due to, among other things, the enhancement of albedo by vol-
canogenic sulphates. In spite of its importance, the rate of $SO_2$ degassing remains a poorly constrained source of sulphate particles through nucleation and/or condensation on pre-existing aerosol (Robock, 2000). Aerosols, in general, play an important role in the radiative budget both directly by scattering and absorbing radiation, and by modifying the cloud microphysics, including their radiative properties and lifetime (Hobbs et al., 1982; Albrecht, 1989; Stevens et al., 1998; Ackerman et al., 2004; Sandu et al., 2009). The volcanic contribution of Cloud Condensation Nuclei (CCN) or Ice-Forming Nuclei (IFN) through gas-
to-particle conversion is estimated to 5-10 $Tg\ yr^{-1}$ (Halmer et al., 2002). This source is comparable to the biogenic sulfides (25 $Tg\ yr^{-1}$) and anthropogenic $SO_2$ (79 $Tg\ yr^{-1}$) contributions (Penner et al., 2001). An objective is to better estimate the volcanic source of CCN and IFN for climate models. It requires (i) to improve the quantification of degassing of volcanoes (mainly $H_2O$, $CO_2$ and $SO_2$) during and between eruptions, and (ii) to parameterize nucleation of aerosols inside the volcanic





plume presenting specific conditions of temperature, humidity and concentration. Real-time, high-frequency quantification for gas emissions during weak to moderate volcanic events can be achieved by integration of multiple ground based techniques (DOAS, Multi-GAS, FTIR) (Conde et al., 2014, and references therein). Specifically, this approach permits a high spatial and temporal resolution of gas emissions, some of them (e.g. $CO_2$, $H_2O$) having a strong atmospheric background. For the second

aspect, Boulon et al. (2011) provided the first observational evidence of the occurrence of aerosol nucleation within a volcanic plume. They also demonstrated that the classical binary nucleation scheme ($H_2SO_4$-$H_2O$) used in meteorological models underestimates observed particle formation rates in volcanic plumes by 7 to 8 orders of magnitude. These results showed that nucleation schemes must be revised for volcanological contexts.

In this context, the french Research National Agency (ANR, http://www.agence-nationale-recherche.fr/) has funded the trans-

disciplinary program named STRAP (Strategie Transdisciplinaire pour Répondre aux Aléas liés aux Panaches Volcaniques) for the period 2015-2019. The objectives of the STRAP experiment is to promote the tracking, sounding and integration of complex physico-chemical processes governing the temporal and spatial evolution of volcanic plumes from the source (high temperature, concentrated) to the regional-scale in which the plume disperses and is subject to dry and wet sedimentation. The STRAP experiment is based upon two observation periods in 2015 (Piton de la Fournaise, France) and 2016 (Etna, Italy).

The 2015 STRAP experiment has been performed on the Piton de la Fournaise (PdF) volcano through a cross-disciplinary collaboration between scientist partners of La Réunion University (LACy; OSU-R/UMS 3365), the Institut de Physique du Globe de Paris (IPGP/OVPF), the Observatory for the air quality of La Réunion (ORA), French laboratories (OPGC: LAMP, LMV from Clermont Ferrand; CNRM/GAME from Toulouse) and international partners (Chalmers University of Technology, Sweden; University of Palermo; Istituto Nazionale di Geofisica e Vulcanologia, Italy). Piton de la Fournaise was selected as

it is a well-known very active volcano (4 eruptions in 2015) and emits lava flows and gas plumes with negligible amount of ash charge (Roult et al., 2012). The gas plumes of PdF have been targeted to (i) estimate day by day the emission rates of $SO_2$, water vapor and $CO_2$; (ii) measure and model the altitude and direction of buoyant plume above the source area; (iii) study the nucleation of aerosols to improve the model parameterization; (iv) study and model the transport, ageing, dry and wet deposition of the volcanic plumes around La Réunion. The objective of this paper is to present an overview of the PdF

STRAP 2015 experiment, a synthesis of the observations and some preliminary results. It is organized as follows. Section 2 describes the morphology and the meteorological conditions of La Réunion island and gives information about the campaign organization. Section 3 describes the peculiarities of the Piton de la Fournaise and the different eruptions that occurred in 2015. Section 4 presents the diurnal evolution and the climatology of the plume during the two main eruption of 2015. The next sections present the instruments, their operating periods and give examples of the main results, respectively for the near-vent

DOAS $SO_2$ emission measurements (section 5), for the LIDAR profile measurement (section 6), for the airborne measurement (section 7) and for the Maïdo observatory measurements (section 8). Section 9 summarizes the main results and draws some preliminary conclusions obtained from the distal experiments.





## 2 Local context

### 2.1 Topography and climate of La Réunion island

La Réunion (-21°, 55°) is a volcanic island located in the southwestern part of the Indian Ocean. The topography of La Réunion island is unique and craggy. Piton des Neiges, located in the north central region of La Réunion, is the highest point on the island and reaches 3,070 meters height. Located in the southeast corner of La Réunion Island, Piton de la Fournaise (2,631 meters) is one of the World's most active volcanoes. Three cirques (Salazie, Mafate and Cilaos) surround the peak of Piton de Neiges (Fig. 1). They have been formed by the volcanic activity which collapsed an original volcanic edifice and were then scoured out by intense erosion (Lenat, 2016).

La Réunion benefits from a tropical climate softened by the breezes of the Indian Ocean. The island is affected by south-easterly trade winds near the ground, and westerlies in the free troposphere. The eastern and western parts of the island are wet and dry, respectively. Clouds develop daily on the summits and on the flanks of the island, with a well-established diurnal cycle (formation in the late morning, dissipation at the end of the afternoon or at the beginning of the night).

### 2.2 Observation means during the STRAP campaign

The Observatoire Volcanologique du Piton de la Fournaise (OVPF/IPGP) managed the monitoring networks on the island, permitting to supervise and follow the volcanic events and to describe their time and space evolution. In the frame of the STRAP program, the OVPF team was in charge of the quantification of volcanic gas emissions close to the vent and of the sampling of solid products. Internationals partners from INGV (Italy) and Chalmers University (Sweden) have contributed to data acquisition and interpretation. OVPF has implemented since its installation four permanent ground based networks: i) imagery, ii) seismological, iii) geodetical and iv) geochemical. Multi spectral satellite imagery, and ground based and drone visible and thermal imagery complement these datasets. The Observatoire de Physique de l'Atmosphère de la Réunion (OPAR) organized the observations of volcanic plumes at larger distance across the island. During the STRAP campaign, the OPAR has coordinated different laboratories such as LACy (Laboratoire de l'Atmosphère et des Cyclones), UMS 3365 from OSU-R (Observatoire des Sciences de l'Univers de la Réunion), LaMP (Laboratoire de Mesures Physiques) and CNRM (Centre National de La Recherche Météorologiques). ORA (Observatoire Réunionnais de l'Air) has also participated by making available their air quality network.

The field campaign has been based on two main measurement sites located near the vent and at the Maïdo observatory (Baray et al., 2013) located at 2200 $m$ altitude and at 43 $km$ northwest of the Piton de La Fournaise (Fig. 1). The gas geochemical network of OVPF groups three techniques: DOAS (Differential Optical Absorption Spectroscopy), Multi-GAS (Multicomponent Gas Analysis System) and an accumulation chamber for $CO_2$ soil flux. Gas data interpretation is constrained by the geochemical analysis of the composition of the eruptive products (naturally or water quenched), performed at LMV (Laboratoire Magmas et Volcans) and IPGP. Time series analysis of gas geochemical data are correlated with geophysical (seismicity, deformation) data series to assess the time and space evolution of the eruptions, the dynamics of the magma and fluids storage zone(s) and to put constraints on their volume and pressure (depth), as well as to make inferences on the rate and timing





of magma and fluid transfer and accumulation at depth (Peltier et al., 2016; Michon et al., 2013). In the framework of the STRAP project, permanent scanning and mobile remote DOAS measurements of the $SO_2$ mass emission rate, in-situ Multi-GAS measurements of the molar concentrations of $H_2O$, $CO_2$, $H_2S$, and $SO_2$ (Table 1) and rock sampling (Table A1) have been performed. The temporal and spatial evolution of lava emissions were constrained by detailed mapping of the lava field

combined with satellite imagery (Coppola et al., submitted). At the Maïdo observatory, gases such as $SO_2$, $NO_x$, $CO$, $O_3$, the size distribution and the extinction of aerosols, and the CCN (Cloud Condensation Nuclei) size distribution are measured continuously. During the STRAP campaign, an AIS (Air Ion Spectrometer) and a nano CPC (Condensation Particle Counter) have been added to estimate the aerosol nucleation. Mobile measurements were made during May and between August and mid-September 2015 to scan the plume evolution. An ultralight aircraft equipped with on-board instruments has measured the

$SO_2$ molar concentration, the particle number and size distribution of aerosols. The ultralight aircraft was located at the air base of Saint-Paul (Fig. 1). The operating range ($\sim$ 1h30) of the instruments was sufficient to reach the area of the Piton de la Fournaise and to make several transects through the $SO_2$ plume. An aerosol LIDAR was installed on a truck coupled with a set of batteries. This flexible system enabled us to move under the aerosol plume in order to measure the aerosol optical properties, the thickness and the horizontal extension of the plume. The area of the truck measurement and the flight trajectory were

planned every day by a briefing using forecasts from the French Meteorological Office (DIROI / Météo-France) and tracking simulations of the dispersion of the volcanic plume.

## 3    The Piton de la Fournaise volcano

### 3.1    Description of the recent activity of Piton de la Fournaise

Piton de la Fournaise is a very active basaltic volcano, the youngest tip of the 5000 $km$ long hot-spot chain that fed the Deccan

Trap flood basalt 65M years ago (e.g. Duncan, 1981). About 97% of the recent eruptive activity occurred in the 5465-2971 years old "Enclos Fouqué" caldera (Fig. 1), a approximately $8 \times 13$ $km^2$ wide structure, which is an uninhabited seaward open natural depression (Villeneuve and Bachèlery, 2006; Michon et al., 2015; Ort et al., 2016). This activity has contributed to the formation of the 400 $m$ high summit cone whose top consists of two craters, the Dolomieu and Bory craters, to the east and west, respectively. According to the location of the eruptive fissures, three types of eruptions have been defined (Peltier et al.,

2009): (i) summit eruptions located inside the summit craters, (ii) proximal eruptions located on the flank of the summit cone along the rift zones, striking N10-25, N160-180, and N120 (Michon et al., 2007, 2015; Bonali et al., 2010), and (iii) distal eruptions located outside of the summit cone, more than 4 $km$ from the summit. According to this classification, eruptions studied in May and August 2015 by the STRAP teams are classified as proximal (Fig. 1).

During the last decades, the volcano erupted at a mean rate of 1 eruption every 9 months, with a mean extruded volume of

lava of 8 million of $m^3$ ($Mm^3$) (Roult et al., 2012). Even if phreatic and phreatomagmatic eruptions have been described in the eighteenth and nineteenth centuries (Peltier et al., 2012; Michon et al., 2013), since the end of the $18^{th}$ century the eruptive activity has mostly consisted of dominantly effusive or weakly explosive eruptions. A characteristic eruption consists of the opening of en-echelon fissures, with activity rapidly focusing on the lowest point of the feeding fissure and forming a small





cone from the accumulation of lava fountaining products, and lava flow effusion. The associated volcanic plumes are often little ash-charged. The last major ash emissions occurred during the major April 2007 event, when the Dolomieu summit crater collapsed, and during which ash deposit thicknesses between less than 1 $mm$ to 15 $mm$ have been reported around the summit craters (Staudacher and Peltier, 2015). During this major historical eruption (210 $Mm^3$ of emitted lava flows), a total $SO_2$ of

230 $kt$ has been released, among which 60 $kt$ have been transformed into $H_2SO_4$, and 27 $kt$ of $SO_2$ and 21 $kt$ of $H_2SO_4$ have been deposited at the surface by dry deposition (Tulet and Villeneuve, 2011). During the shortest lived eruptions, lava and gas emission rates are linearly correlated and both decline quickly after an initial intense phase (Hibert et al., 2015). Periods of intense eruptive activity (e.g. 1972-1992; 1998-2010; mid 2014-today) alternate with more or less long periods of rest (e.g. 1966-1972; 1992-1998; 2011-mid 2014; (Peltier et al., 2009; Roult et al., 2012). During the 1998-2010 period of sustained

activity, Piton de la Fournaise volcano produced 2.6 eruptions per year, on average, spanning a very large range in emitted volume of lava between < 1 $Mm^3$ and 210 $Mm^3$ (Peltier et al., 2009; Staudacher et al., 2009; Roult et al., 2012). In 2007, a major pulse of deep magma triggered the expulsion of 210 $Mm^3$ of magma mostly stored in the shallow part of the volcano plumbing system. It provoked the collapse of the Dolomieu summit caldera to a depth of 320±20 $m$ (Staudacher et al., 2009; Di Muro et al., 2014). From 2007 to 2010, the volcanic activity was relatively weak, with several short lived, small volume

eruptions and intrusions mostly located close to the central cone (Peltier et al., 2010; Roult et al., 2012; Staudacher et al., 2015). The weak 2008-2010 eruptive activity was followed by a relatively long phase of quiescence of 41 months (Feb. 2011-June 2014) before a fast reawakening in June 2014 (Peltier et al., 2016). During the phase of quiescence, the volcano deflated. Daily micro-seismicity was limited to a few shallow volcano-tectonic events, and summit intra-caldera fumaroles emitted low temperature $S$ and $CO_2$ poor water vapor-dominated fluids.

## 3.2   Description of the eruptions during the STRAP period (2014-2015)

In 2014-2015, the OVPF recorded an increasing rate of activity in terms of deformation, seismicity, gas emissions and eruption rate culminating in five eruptions Peltier et al. (2016). Like most eruptions of the recent historical activity, they all occurred inside the Enclos Fouqué caldera. This last sequence of eruptions showed a progressive increase in the output rates (Coppola et al., submitted). The different eruptive events include: 20-21 June 2014 (south-southeast of the Dolomieu crater; 0.4±0.2

$Mm^3$ of erupted products); 4-15 February 2015 (west-southwest of the Bory crater; 1.5±0.2 $Mm^3$ of erupted products); 17-30 May 2015 (to the south-southeast; 4.6±0.6 $Mm^3$ of erupted products); 31 July - 2 August 2 2015 (to the north; 2±0.3 $Mm^3$ of erupted products); and August 24-October 31, 2015 (to the west-southwest; 35.7±3 $Mm^3$ of erupted products in 3 successive stages: 24 August - 18 October, 22-24 October, and 29-31 October). Renewal of eruptive activity in 2014-2015, after 41 months of quiescence and deflation, was associated with long-term continuous edifice inflation measured by GNSS (Global

Navigation Satellite System). Inflation started on June 9, 2014, and its rate progressively increased through 2015. The time evolution of monitoring parameters has revealed that the volcano reawakening was associated with continuous pressurization of the shallowest parts of its plumbing system, triggered by progressive upwards transfer of magma from greater depth (Peltier et al., 2016). Peltier et al. (2016) conclude that, in 2015, both a pulse in the deep refilling, as well as the perturbation of



## 4 Transport modelling of volcanic tracer

### 4.1 Description of the FLEXPART simulation model

The Flexfire forecasting tool is based on the coupling between the FLEXPART Lagrangian particle dispersion model (Stohl et al., 2005) and the AROME mesoscale weather forecasting model (Seity et al., 2011). To calculate Lagrangian trajectories, we used a version of the FLEXPART Lagrangian particle dispersion model modified to be coupled with AROME output. The model uses the horizontal grid coordinates from AROME, and uses a terrain following Cartesian coordinates for vertical levels. Because meteorological parameters such as wind, humidity and temperature were available on 12 Cartesian terrain following

vertical levels up to 3000 $m$ altitude and on a dozen of pressure levels up to 100 hPa, a vertical interpolation for the different meteorological fields available on the pressure levels has to be performed for altitudes higher than 3000 $m$ agl (above ground level). The Hanna turbulent scheme (Hanna, 1982) was used to represent turbulent mixing in the boundary layer. Parameters used in the Hanna scheme are calculated internally by FLEXPART (e.g., PBL mixing height, friction velocity, etc.). To run the model with an accurate representation of the turbulent mixing in the PBL, the time step in FLEXPART is calculated depending

on the values of a Lagrangian timescale, vertical velocity and mixing height at each trajectory position. An upper limit was fixed at 180 $s$ for horizontal mixing and 36 $s$ for vertical mixing. Since the Eulerian meteorological fields were available at 0.025×0.025 ° resolution, the mesoscale meandering term in the FLEXPART Lagrangian trajectories was removed. During the field campaign, forward FLEXPART trajectories were simulated over 36 $h$ using the AROME forecasts. Each trajectory carried an equal mass of a passive tracer based on an arbitrary surface flux. Overall, 40000 trajectories were released per FLEXPART

forecasts. These forecasting were used day-by-day to manage the strategy of the observations (e.g. microlight aircraft flight plan, LIDAR location, portable DOAS measurements).

### 4.2 Simulated evolution of a volcanic tracer load during STRAP

After the field experiment, FLEXPART was re-run to simulate the atmospheric transport for the main eruptive time periods as a post processing tool. A total of 1.5 million trajectories were used every day to simulate the atmospheric transport of

volcanic plumes. The tracer used in the FLEXPART simulation is assumed passive to avoid low biases due to uncertainties in the precipitation fields from AROME. This tracer was dedicated to evaluate the transport and dispersion of the volcanic $SO_2$. The temporal mean of the tracer load was modelled for the two main eruption periods of May 2015 and August-to-October 2015. Figures 2 and 3 show the probability of occurrence of the tracer load integrated over all model levels and normalized at 00, 06, 12 and 18 UTC (local time is UTC+4), respectively for the two main periods. The normalization was made by an

average for each hour over the simulation period to model the diurnal evolution of the occurrence of the tracer load. Indeed, the emission area is pointed out with a value normalized to 1 (100% of presence of the tracer plume above the vent). It also



represents the average of dilution of the tracer load in comparison to a load of tracer imposed at 1 at the vicinity of the vent corresponding to the surface of the AROME grid.

During the May eruption period (Fig. 2 ), the tracer plume was dispersed due to an important variability of the wind field. A diurnal evolution of the tracer load is modelled showing a strong influence of the evolution of the boundary layer and the local

circulation associated to the steep topography of the Island. At night and in the morning (18 UTC to 06 UTC), one can observe local maxima of the tracer load simulated to the west (Valley of the Rivière des Ramparts), to the north-west (Tampon-Bourg-Murat) and to the north (Plaine des Palmistes) of the vent (Fig. 1). These maxima characterized the presence of potential zones of $SO_2$ accumulation during night formed by a downwind density current generated by the difference of density between the summit and the valleys. These results are consistent with visual observation from the ultralight aircraft showing a bluish plume

over the Cilaos caldeira, the Tampon area and the Ramparts valley which is typical of sulfur presence. At 06 UTC (morning), the boundary layer thickness has increased and the tracer was modelled above the surface generally between 2500 and 3000 $m$ asl. As a consequence, the occurrence of the tracer load up to 2% is more homogeneously simulated from the south-west to the east of the vent. The mean distance of the 1% tracer load presence does not exceed 20 $km$ from the vent. A strong plume occurrence (load up to 20%) was modelled at 06 UTC north of the vent above the Plaine des Palmistes. In the afternoon (12

UTC), the altitude of the plume was generally simulated up to 3000 $m$ asl and was less influenced by the topography. At this altitude horizontal winds are stronger and vertically sheared due to the Trade wind inversion. This can be observed on Figure 2 where the tracer plume is advected over higher distances. Four zones up to 2% of tracer load is simulated to the west, north, north-east and east of the vent and reach 50 $km$ of distance. This location of the tracer plume is not typical in regards to the mean regime of the Trade winds over La Réunion island (Fig. 1). It also reveals that the transport of gas emitted in the caldera

of the PdF is strongly influenced by the local circulation.

Figure 3 gives the occurrence of the tracer load between 24 August to 31 October modelled by FLEXPART. The field is quite different from the one simulated in May, indicating that weather conditions have changed between the two periods. Overall, the sulphur load is simulated west of the vent and, and in a lesser extent at the north. During the night (18 and 00 UTC) a maxima of the tracer presence (up to 20%) is modelled west of the volcano above the Rivière des Ramparts. The field of

the tracer concentration (not shown), has indicated that this local maximum is due to an accumulation of nocturnal pollution trapped within this valley. In the morning (06 UTC) the distribution of the tracer load occurrence is close to that simulated at 00 UTC. However, the field is more homogeneous and with a higher horizontal extension showing that the boundary layer thickness have increased. The tracer plume is located at a higher altitude which makes it less influenced by the topography. In the afternoon (12 UTC), this process is more marked and the fields are homogeneous around the vent except at the south-east

where the presence of the tracer is low. The presence of the tracer load is mainly modelled to the west and south-west from the vent above the area of the city of Saint-Pierre. This well-marked distribution shows that during the period, the Trade winds were essentially zonally oriented near the vent. According to the FLEXPART model, the number of days where the plume of the tracer could directly interact with the Maïdo observatory is less important than during the May period (not shown).



## 5 Source emissions at the volcanic vent

The output budget of the main components of the vent gas emissions has been quantified by integrating the $SO_2$ fluxes (DOAS) with the ratios of the main species ($H_2O/SO_2$, $CO_2/SO_2$), measured in situ close to the eruptive vent by portable MultiGAS. The three permanent scanning DOAS stations of OVPF (Partage, north; Enclos, west; Bert, south) are installed close to the rim of Enclos Fouqué caldera and have worked continuously during the whole August 2015 eruption. Details of the instrument and evaluation procedure are given in Galle et al. (2010) . The three stations have permitted to acquire up to 115 scans of the plume every day (on average 34 scans every day or 1 complete plume scan per 13 minutes) and to estimate $SO_2$ column density, plume height and dispersion direction (Fig. 4), as well as the total $SO_2$ emission rate. Poor quality datasets have been acquired on 13 September and 03 October, due to rainy weather conditions. Trasects of the plume were periodically acquired by portable DOAS in order to validate data acquired by portable stations or to guarantee a better coverage of unusual plume geometries. For both DOAS datasets, $SO_2$ fluxes have been obtained by integrating the column densities across the plume cross section and scaling with wind speed data kindly provided by the Météo-France station located at Bellecombe, in between the Partage and Enclos stations. Wind data are acquired hourly and on a mast at $10\ m$ agl. Plume geometry (height, direction) and $SO_2$ emission rates ($ton\ day^{-1}$) are presented on Figures 4 and 5. They have been compared along with daily rain data measured at the OVPF ChateauFort station, near the south-east base of the summit cone, to give an indication of the quality of the results since rainy conditions are detrimental. DOAS sessions are acquired with a high rate. However it is important to keep in mind that, on average, they correspond to 1/3 of the total daytime, because the DOAS stations use skylight as the source of radiation. Extrapolation of the average $SO_2$ fluxes measured during this time window to the whole day duration may cause an observational bias. However, the average pattern of wind directions during night-time is the opposite to that on day-time, and would often escape observation by the DOAS network. Nevertheless, interesting comparison with independent datasets is still possible. The plume height and direction were obtained by triangulation from the scans of two DOAS stations (see Galle et al. (2010) for details). Plume directions were mostly blowing south-westwards during daytime (57±25 °, n=780), consistently with FLEXPART simulations (Fig. 4). Plume directions are only partly consistent with wind directions measured close to the ground at the Bellecombe station, possibly reflecting a wind rotation with elevations, since the plume height was up to 4-5 $km$ above sea level. Directions are increasingly consistent with increasing plume height and increasing wind speed (Fig. 4). The plume height varies during the eruption, reflecting the eruptive intensity and modulation by local meteorological conditions. It was consistently above the level of the volcano summit and on several occasions above the boundary layer (3063±332 $m$ asl, n=780).

The $SO_2$ flux is reported on Figure 5 as daily mean values and standard errors. The 20 May, the mean of $SO_2$ emission measured by DOAS has reached 1870 $t\ day^{-1}$ and corresponds to the maximum observed during the whole 2015. Two other phases of high mean emissions at 1000 $t\ day^{-1}$ and 1840 $t\ day^{-1}$ have been measured respectively at the begining of September and at mid-October. Individual scan measurements were validated by selecting only completely sampled plumes. Under this geometrical condition, the plume is well elevated above the ground level and its transport corresponds to a given range of wind directions for each station. The daily average was computed by weighting each measurement by its uncertainty



and duration. The individual scan measurements are presented in Figure 6 May and for the August-October eruption. As the August eruption started at night-time, the DOAS network partly missed the initial, intense, but short lived phase of high output rate on 24 August. During this phase, $km$-long eruptive fractures opened at the south-west feet of the summit cone. Opening of a small short-lived vent on the north-western flank of the cone showed that initially the feeder dyke crossed the entire summit

cone. Maximum lava output rate on 25 August estimated by $SO_2$ fluxes is still in the range 24-37 $m^3\ s^{-1}$ (depending on the assumed initial amount of dissolved $SO_2$). Volcanic activity evolved rapidly into a single emitting source and the eruption focused to build a single and large eruptive cone, feeding a broad lava field. The temporal trend in $SO_2$ emission permits to identify the following phases of the August 2015 eruption:

– Phase 1 (24/08 - 12/09): trend of progressive decrease in $SO_2$ emissions

– Phase 2 (13/09 - 18/10): trend of accelerating increase in $SO_2$ emissions

– Phase 3 (20/10 - 01/11): two important discrete pulses in $SO_2$ emissions

Peaks in $SO_2$ fluxes where highest (in the range 5-6 $kt\ day^{-1}$) at the beginning of the eruption (25-27 August) and in the period 25 September to 30 October. Interestingly a peak in $SO_2$ emissions (up to 6.8 $kt\ day^{-1}$) was also recorded at the end of phase 1, in the morning of 10 September (between 3:51 and 5:40 UTC). Preliminary observations on the eruptive products

suggest that this peak in $SO_2$ might have heralded a change in magma composition from phase 1 to phase 2 (Coppola et al., submitted). During most of the eruption, $SO_2$ fluxes have been lower than 1.5-2 $kt\ day^{-1}$. In the 11-17 September period, very low $SO_2$ emission rates (<300 and 600 $ton\ day^{-1}$, respectively) were estimated. Possibly, these weak emissions correspond to a phase of low altitude of the gas plume, mostly confined inside the Enclos Fouqué caldera, during the end of phase 1. Weak emissions in the 01-06 October period might be related to a phase of intense rain, which might have partly scavenged $SO_2$

from the atmosphere and introduced radiative transfer effects on the flux measurements. NOVAC data suggest that magma and gas output rate accelerated during the second part of phase 2, i.e. between 8 and 18 October. According to Coppola et al. (submitted), this change corresponds to the emission of more gas rich primitive magmas and to an increase in the lava emission rate. Interestingly, the strong increase in mass flow rate on 15-17 October is associated with a strong increase in gas plume height (up to 5.1 $km$ above sea level) over the period 16-18 of October.

During the pulsating final phase (phase 3) of the eruption, $SO_2$ emissions are strongest during the second and final eruptive pulse (Fig. 5 and 6). During the first eruptive pulse of phase 3, fast lava propagation triggered a major fire and aerosol/ash emissions from the southern walls of the Enclos Fouqué caldera. An attempt of estimating lava output rate and total emitted volume of lava from $SO_2$ emissions has been performed using the procedure calibrated by Hibert et al. (2015) on the low intensity January 2010 eruption. This procedure requires some assumptions on the pre-eruptive $S$ content, $S$ speciation and

phase partitioning, and on the density of the emitted lava. This approach is clearly an oversimplification for a complex long lasting eruption like that of August 2010. For instance, the chemical zoning potentially translates in highly changing initial $SO_2$ contents and it produces a high error in the estimate. Using Hibert et al. (2015)'s approach and average daily $SO_2$ emission rates a potential lava bulk volume of 21-39 $Mm^3$ is obtained. This value is similar but on average lower than that





estimated (by photogrammetry 36±3 $Mm^3$ see Peltier et al. (2016) and by satellite data 45±15 $Mm^3$ see Coppola et al. (submitted)). As discussed by Coppola et al. (submitted), pre-eruptive $S$ degassing of shallow stored magma might explain this discrepancy. In the proximal area and close to the eruptive vents, regular survey and visible and thermal imagery have permitted to follow the time and space evolution of eruptive dynamics and to constrain the evolution of the degassing source

(Fig. 7). In situ analysis of gas emissions have been performed using a mobile MultiGAS device (5 sets of measurements in May and 12 in August-October) during both eruptions targeted by the STRAP project. A MultiGAS portable station has been installed or carried downwind and close to the eruptive vent (at a distance of tens to hundreds of meters from the degassing sources) for acquisition times of 15-96 minutes (0.1 $Hz$ sampling rate). On 19 May the measurements were performed from an helicopter during a flight close to the main high temperature degassing source (Fig. 7). The purpose of the experiment was

to measure the relative molar concentration of the main components of the gas phase emitted by several sources (eruptive vent, fumaroles and lavas). The MultiGAS device integrates a dual beam IR spectrometer ($CO_2$), electrochemical sensors ($H_2S$ and $SO_2$), an hygrometer, and pressure and temperature sensors. $SO_2$ concentration up to 95 $ppm$ (95.000 $ppb$) were recorded above the vent. Water is recalculated from hygrometric measurements. Subtraction of the atmospheric background permits the quantification of the elemental molar ratios (e.g. $H_2O/SO_2$, $CO_2/SO_2$ molar ratios) in the volcanic emissions.

Correlation of these ratios with the $SO_2$ fluxes measured by DOAS will permit to estimate the syn-eruptive fluxes of $H_2O$ and $CO_2$ released by the eruptive vent(s). Preliminary data treatment confirms previous findings at Piton the la Fournaise that concentrated, high temperature gases are water dominated, with variably high $H_2O/CO_2$ molar ratios (50-240) and that sulphur is the second species before carbon ($CO_2/SO_2$ molar ratio < 0.6). A more complete description of the MultiGAS system used in our experiments can be found in Aiuppa et al. (2009). In May and August 2015, the OVPF team sampled 10

and 50 rock samples for geochemical analyses, respectively (Table A1). Bulk rock, crystal and melt inclusion composition are under study to constrain the pre-eruptive storage conditions of the magma (pressure, temperature, gas content and composition) and to identify the possible involvement of several magma sources. Preliminary data suggest that the May eruption was fed by a single and relatively homogeneous magmatic source, while the long lasting August eruption involved at least two magma sources during phases 1 and 2 respectively (Di Muro et al., 2016; Coppola et al., submitted). According to Coppola et al.

(submitted) the 2014-2015 sequence of eruptions provoked the progressive draining of the upper and chemically zoned portion of the volcano feeding system, triggered by the ascent of new deep inputs. We want to stress here that this is the first time that such a complete and detailed geochemical dataset is acquired at Piton de la Fournaise, both during short lived (May 2015) and long lasting (August-October 2015) eruptions.

## 6   Volcanic plume height and structure, aerosols optical properties

We used a mobile aerosols LIDAR in synergy with a handheld sun photometer to measure the volcanic plume height and to retrieve optical properties of the encountered aerosols. The LIDAR system used in this study is a LEOSPHERE ®ALS450 based on a Nd:Yag laser producing pulses with a mean energy of 16 $mJ$ at 355 $nm$ and a frequency of 20 $Hz$. Lidar measurements have been averaged over 2 $min$ with a vertical resolution of 15 $m$. The LIDAR profiles enable to retrieve atmospheric





structures (boundary layer heights, aerosol layers and clouds) and optical properties (LIDAR ratio (LR) and extinction coefficient profiles) in synergy with sun photometer measurements. It is particularly well-adapted to planetary boundary layer (PBL) studies, thanks to its full overlap height reached at $150\ m$. A more complete description of the LIDAR and its instrumental features can be found in Duflot et al. (2011). For this campaign, the system was installed in an air conditioned box adapted for

use in severe conditions. Depending on the FLEXPART forecast simulations (section 4), it was either set at a fixed position at the OVPF site in Bourg Murat (-21.12°, 55.34 °; $1560\ m$ asl), or attached on a pick up platform to perform mobile observations of the volcanic plume. In this last case, the energy was supplied by 6 batteries connected to a power inverter, which gave approximately 3 hour autonomy to the system.

Aerosol optical thickness (AOT) measurements were performed in clear-sky condition using a MICROTOPS II Sun photometer

instrument (Solar Light, Inc.). The instrument field of view is about 1°. The AOT is measured at five wavelengths (440, 500, 675, 870 and $1020\ nm$). The instrument was calibrated at the NASA Goddard Space Flight Center against the AERONET reference CIMEL Sun/sky radiometer. The data presented here have been quality and cloud screened following the methodology of Smirnov et al. (2000) and Knobelspiesse et al. (2003) and the mean uncertainty on the AOT measurements equals 0.015 (Pietras et al., 2016). The AOT at the LIDAR wavelength of $355\ nm$ ($AOT_{355}$) was calculated from $AOT_{440}$ using the

Ångström exponent (Ångström, 1964) between 440 and $675\ nm$. The uncertainty on the retrieved $AOT_{355}$ has been computed following the approach showed by Hamonou et al. (1999).

The synergetic approach between LIDAR and sun photometer measurements to calibrate the LIDAR system, to retrieve aerosols optical properties, and to evaluate the uncertainties can be found in Duflot et al. (2011). It is noteworthy that this method gives access to a height-independent LR value. In our case, the contributions of the aerosols trapped in the PBL (sea

salts, aerosols originating from anthropogenic activities such as fossil fuel combustion) and of the volcanic aerosols can not be set apart. The aerosols optical properties given hereafter are therefore valid for the encountered mixtures of aerosols. On 21 May 2015, the plume was forecasted by FLEXPART to be located south-west of the volcano (Fig. 8). We therefore performed measurements between Saint Pierre and the OVPF site in Bourg Murat in order to cross the plume with our mobile LIDAR system. Figure 8 shows the time series of the backscattered signal along the track. The plume is clearly visible and seems to

flow along the relief with a quasi-constant vertical extension from 0 to $1200\ m$ agl (the blank period was due to a default of the system). The high values of the backscattered signal observed at the track's start are plausibly due to a mixing of volcanic and marine aerosols, these latter increasing notably the aerosol loading close to the coast. Figure 8 shows the extinction profile retrieved using simultaneous sun photometer measurements at 6:14 UTC (vertical black dashed line on Figure 8). The measured $AOT_{355}$ and Ångström (Å) coefficient between 500 and $675\ nm$ are 0.20±0.05 and 0.91±0.14, respectively, and the retrieved

LR at $355\ nm$ ($LR_{355}$) is 63±16 $sr$. The extinction profile shows a vertical extension of the plume up to $2500\ m$ asl with a maximum extinction value of $0.19\ km^{-1}$ reached at $2200\ m$ asl and a second maximum value of $0.16\ km^{-1}$ reached at 1100 $m$ asl. The complex structure of this observed aerosols extinction profile is probably due to the mixing of sea salt and anthropogenic pollution-loaded coastal air masses lifted up by the sea breeze (Lesouëf et al., 2011) with volcanic aerosols-loaded air masses flowing down the slope. In this case, each aerosol type (oceanic, anthropogenic, and volcanic) brings its own contri-

bution to the observed Å and $LR_{355}$ values, which reflect this aerosols mixture (see Cattrall et al. (2005) for typical values of





optical aerosols parameters observed for oceanic and anthropogenic urban aerosols). In addition, the processes influencing the volcanic aerosols growth (such as coagulation) could impact the observed size of the encountered aerosols. Figure 8 also shows the tracer profile simulated by FLEXPART at 6 UTC (10 $h$ local time) at the same location. Note that this modeled profile only allows to assess the ability of the model to simulate the tracer vertical distribution, not the resulting extinction profile. In other

words, the simulated distribution can only be compared qualitatively to the observed one. Moreover, the model provides hourly outputs, which prevents us from comparing the simulated distribution at the exact same time as the observations. Finally, the model only simulates the volcanic tracer distribution, and does not take into account other types of aerosols (marine and anthropogenic in our case) possibly encountered and mixed with the volcanic ones. Having these caveats in mind, one can see that the model shows the same vertical plume extension up to 2500 $m$ asl, and the same two tracer loading peaks at 1000 and

2000 $m$ asl, which are 200 $m$ below the observed ones. However, opposite to the observations, a higher loading is simulated at 1000 $m$ than at 2000 $m$. On 2 September 2015, the plume was forecasted to be located south-west and west of the volcano and the LIDAR was consequently installed at the OVPF (16 $km$ west from the volcanic vent, Fig. 9). On Figure 9, the aerosols layer is clearly visible between approximately 1600 and 3000 $m$ asl (0 and 1500 $m$ agl) with a highly loaded aerosols layer up to 2100 $m$ asl (500 $m$ agl). One can also notice a fresh crossover of aerosols plume starting at 11 UTC, credibly coming

directly from the vent. Figure 9 shows the extinction profile retrieved using simultaneous sun photometer measurements during this fresh aerosols crossover at 13:10 UTC (vertical black dashed line on Fig. 9). The measured $AOT_{355}$ and Å are 0.15±0.04 and 1.26±0.19, respectively, and the retrieved $LR_{355}$ equals 42±10 $sr$. The extinction profile locates the plume top at 3000 $m$ asl (1400 $m$ agl) and shows the highest aerosol loading below 2400 $m$ asl (800 $m$ agl) with a maximum value of 0.20 $km^{-1}$ reached at 2000 $m$ asl (400 $m$ agl). The relatively high Å value (1.26±0.19) measured in this case indicates the presence of

small particles and the retrieved $LR_{355}$ value (42±10 $sr$) is in agreement with LR values retrieved by Pisani et al. (2012) for freshly emitted volcanic aerosols (30-45 $sr$ at 532 $nm$ at about 7 $km$ from the Etna summit crater). These observations are therefore in agreement with the hypothesis of a fresh volcanic plume partially made of sulfuric acid droplets. A more detailed analysis of the various $AOT_{355}$, Å, $LR_{355}$ and extinction profiles observed and retrieved during the 2015 Piton de la Fournaise's eruptions will be the subject of a dedicated study. The tracer profile simulated by FLEXPART (Fig. 9) shows a plume located

between 2600 and 3000 $m$ asl (1000 and 1600 $m$ agl). This poor agreement between the simulation and the observation in this case is also visible on Figure 9 which shows the OVPF site located at the very edge of the simulated plume.

## 7    Microlight aircraft measurements

During the eruptive events of the PdF, in-situ airborne measurements were performed to characterize both the gas and aerosols emitted from the volcano and the spatial distribution of the volcanic plumes. Details on the flights are compiled in Table 2.

The microlight aircraft (FK9 ELA) equipped with onboard instruments conducted 18 flights from 19 May to 18 September 2015 over La Réunion. The eruptive events were probed extensively, including horizontal and vertical volcanic plume structures. The aircraft conducted measurements up to an altitude of 6 $km$ asl and was equipped with a piston engine (ROTAX 912) that allowed flying inside the plume at a minimum distance of 2 $km$ from the eruption vent. Together with the Multi-



Gas measurements onboard the helicopter (section 5), these measurements appear to be the most direct in-situ measurements reported until now in the eruption plume of the PdF. The aircraft payload included meteorological sensors for temperature, pressure and relative humidity. Gas phase measurements of sulphur dioxide ($SO_2$) were made using a UV Fluorescence $SO_2$ Analyzer (Teledyne, model T100), which relies on pulsed fluorescence. The integrated number concentration of particles larger

than 7 $nm$ in diameter was measured using a butanol-based condensation particle counter (MCPC, Brechtel model 1720). A customised version of MetOne OPC (Profiler 212 without heater) was used to measure aerosol size distributions between 500 $nm$ and 10 $\mu m$ in 8 size classes. The particle loaded air was fed through an isokinetic inlet into the CPC and the OPC. Data were recorded during the flights by a processor-based Mbed microprocessor board and stored on microSD card. Figures 10 and 11 show examples of the evolution of $SO_2$ and aerosol concentrations measured along the aircraft trajectories on 1 and

2 September 2015. The results of the trajectory analysis show that trade winds turned the volcanic aerosol plume towards the northwest of La Réunion during these flights. On 1 September 2015, the aircraft entered inside the plume at an altitude of around 2.5 $km$ asl in agreement with DOAS scans. Outside the plume, low concentrations of $SO_2$ ($\sim 10$ $ppb$) were detected, most likely due to the remote location of La Réunion situated far from major anthropogenic influences. By entering the volcanic aerosol plumes, the concentration of $SO_2$ increased dramatically up to 2004 $ppb$. The particle number concentration

reached 65,250 $cm^{-3}$ within the volcanic cloud and was highly correlated to $SO_2$ concentration. Meanwhile, there was a moderate increase in coarse particle concentrations ($D_p > 500$ $nm$). It is very likely that the particles in the volcanic plume were generated by oxidation of volcanic $SO_2$ and subsequent particle nucleation or by condensation of volatile compounds onto pre-existing fine particles ($D_p < 500$ $nm$). The $SO_2$ and submicronic aerosols observed by the aircraft were concentrated in a narrow well-defined plume of about 14 $km$ of horizontal width. The dispersion of the volcanic plume simulated by the

FLEXPART model is compared with in-situ airborne observations in Figure 10. $SO_2$ was carried out as a tracer for volcanic emissions and plume transport in the dispersion model. The simulations represent well the main flow directions of the volcanic plume and the pronounced $SO_2$ concentrations to the east of the vent that could be observed by in-situ measurements. The good correspondence between measurements and simulations suggests that the plume transport is well characterized by the model. The potential of retrieving plume trajectories in combination with in-situ measurements provides relevant information

to explore tranformations inside the volcanic plume as the plume ages. // On the next day, the width of the aerosol plume was determined during the climb phase of the aircraft between 04:26 and 04:43 UTC. A 600 $m$ thick aerosol plume could be identified at an altitude around 2300 $m$ asl. Along this flight, the aircraft entered inside the plume several times at a distance of 2-6 $km$ away from the vent. Peaks of $SO_2$ and particle number concentrations up to 804 $ppb$ and 43,450 $cm^{-3}$ respectively were measured during the plume traverses. The ratio between the $SO_2$ and the particle number concentration observed is not

constant when the aircraft is above the boundary layer (i.e. above 1000 $m$ asl at this period of the day). This clearly shows that the process of oxidation of the volcanic $SO_2$ into $H_2SO_4$, the nucleation of $H_2SO_4$ and the coagulation of fine particles are complex and need to be analysed in detail. It will be the subject of a dedicated study. The particle size distributions of coarse particles in the volcanic aerosol plume had a similar pattern than in the aerosol-free air masses at the same altitude (around 2.6 $km$ asl) and do not indicate the exclusive presence of volcanic particles. The volcanic particles were probably mixed with





other particles types such as anthropogenic particles emitted from pollution transport (Lesouëf et al., 2011) or locally produced biogenic compounds (Duflot et al., submitted).

## 8    Maïdo measurements

The aerosol and ion size distributions, starting from the nanometer sizes, were measured continuously at the Maïdo observatory. The objective was to of characterize and quantify New Particle Formation (NPF) events when the volcanic plume is crossing the site, and to relate them to the presence of sulphuric acid formed within the plume. The size distribution of the 10-500 $nm$ aerosol particles were measured with a Differential Mobility Particle Sizer (DMPS) while the 0.8-42 $nm$ size distribution of the naturally charged fraction of aerosols were measured with an Air Ion Spectrometer (AIS). Here we use ion size distributions below 10 $nm$ as tracers for the presence of neutral sub-10 $nm$ particles that could not be detected directly. A proxy of the sulphuric acid concentration was calculated from the concentration of $SO_2$, measured with certified ASQUA instrument and radiation measured with pyranometer. The DMPS is custom-built with a TSI-type Differential Mobility Analyzer (DMA) operating in closed loop and a Condensation Particle Counter (CPC, TSI model 3010). The quality of the DMPS measurements was checked for flow rates and RH according to the ACTRIS recommendations (Wiedensohler et al., 2012). DMPS measurements were performed down a Whole Air Inlet of a higher size cut of 25 $\mu m$ (under average wind speed conditions of 4 $m\ s^{-1}$). The AIS is developed by Airel Ltd (Estonia), for in-situ high time resolution measurements of ions and charged particles (Mäkelä et al., 1996). The device consists of two DMA arranged in parallel which allows the simultaneous measurement of both negatively and positively charged particles (Gagné et al., 2011). The AIS was directly connected to ambient air through a 30 $cm$ copper tubing of 2.5 $cm$ in diameter, for limiting cluster ion losses along the sampling line. The AIS measurements allow for the calculation of the growth rate (Hirsikko et al., 2007) of newly formed particle from their lowest sizes. Both instruments operated simultaneously over the three eruptive periods from 9 May to 18 October2015 (Fig. 12), together with $SO_2$ measurements operated by ORA (type Thermo Scientific 43i).

The evolution of the $SO_2$ concentration measured at the Maïdo observatory during the year 2015 is shown in Figure 13. The impact of the volcanic plume is clearly visible during a few days in May when the $SO_2$ measurements started, and frequently during the September-October period. Interestingly, the impact is stronger during the short lived small volume eruption of May than during the large long lasting eruption of August-October. Moreover, the impact of August-October eruption is larger during its weaker phase (September) than during the intense phase of activity recorder later on. We chose to focus on two contrasted 2-days periods in term of $SO_2$ concentrations, to investigate the NPF process within the volcanic plume. On the 20 and 21 May, $SO_2$ concentration reaches 433.5 and 288.1 $ppb$ respectively, while on 1 and 2 September, concentrations of $SO_2$ are 81.3 and 7.9 $ppb$, respectively (Fig. 13).

Bulk aerosol samples were collected only under free tropospheric conditions (between 18 and 1 UTC) on a weekly basis. Filter samples (0.5 $\mu m$ pore size diameter Teflon) were analyzed for the main anions and cations by Ion Chromatography. Figure 13 shows the temporal variation of $SO_2$ and sulfate collected on aerosol filter. When the Piton de la Fournaise was active ($SO_2$ showing large concentrations), sulfate showed clear peak concentrations up to 2 to 3 $\mu g\ m^{-3}$ in September 2015. We have





averaged $SO_2$ data to match the sampling period of filter aerosols (between 18 and 1 UTC). The calculated molecular ratio $SO_2/SO_4^{2-}$ during night gives 534, 23 and 8 on 20th May, 1 September and 1 October, respectively. The strong variability of this ratio needs to be further examined. It could be attributable to the strong seasonal variability of UV at La Réunion and thus to the oxidant concentration such as hydroxyl radical. The daily UV index maximum measured in clear sky by the Bentham

spectrometer (Brogniez et al., 2016) at the OPAR site of Saint-Denis (80 $m$ asl) goes in the way of this assumption. The evolution of UV index measured was 6.68 on 20 May, 8.35 on 1 September and 11.26 on 12 October 2015.

For these two distinct periods, examples of the evolution of the ion and aerosol size distribution issued from the combination of the AIS and DMPS measurements are shown, for 20-21 May and 1-2 September, respectively (Fig. 14). The smallest ions (1-10 $nm$) concentration increases in the early hours of the morning (around 6 UTC) in the size distribution. The appearance

of the cluster ions is rapidly followed by their fast growth from 10 to 50 $nm$ on 1-2 September, and 100 $nm$ on 20-21 May that can be observed from the DMPS size distribution. The concentration of newly formed and grown particles is also significantly higher for the May period. The aerosol number concentration in the 10-20 $nm$ size range (Fig. 15) reaches $4 \times 10^4 cm^{-3}$ and $2.5 \times 10^5 cm^{-3}$ on the 20 and 21 of May respectively, and $6 \times 10^4$ and $5 \times 10^4 cm^{-3}$ on the 1 and 2 of September respectively. The order of magnitude of the aerosol number concentration observed during these periods is significantly higher than ob-

served out of the eruption period. Daily maximum concentration at the Maïdo is about $3 \times 10^3 cm^{-3}$ in normal condition and can reach $2 \times 10^4 cm^{-3}$ whene the observatory is under the influence of anthrogenic pollution. These differences show that most of the particles measured on 20-21 May and on 1-2 September have a volcanic origin. The sudden decrease in particle concentration between 16 and 18 UTC to low night time concentration is likely due to a shift of air mass origin due to nocturnal stratification and measurement of the free troposphere. The morning advection of a relatively wide range of ultrafine particles

to the Maïdo station indicates that nucleation and early growth takes place already at the vicinity of the crater, and continues within the plume at least up to the Maïdo station. The common assumption in the scientific community is that the sulphuric acid is the major precursor gas in nucleation processes, due to its low saturated vapour pressure under classical atmospheric temperatures (Marti et al., 1997). This particle is predominantly produced by the oxidation of $SO_2$. We estimated the $H_2SO_4$ concentration from the $SO_2$ ($ppb$) concentration, the radiation ($W\ m^{-2}$), relative humidity (%) and condensation sink (Pirjola

et al., 1999) using the empirical proxy proposed by Mikkonen et al. (2011). Unlike other parameters, the $SO_2$ concentration variation is not periodic because it depends whether the volcanic plume arrives at the station. Except the 20 May, when early morning $SO_2$ data are missing, we observe that the $H_2SO_4$ concentration rises as the same time than the ion concentration increases (around 6 UTC) and approximately two hours before the 10-20 $nm$ particle concentrations increase for all cases. The time delay between the increase in $H_2SO_4$ and in the particle concentration is likely due to the time for new 1 $nm$ clusters

to grow into the 10-20 $nm$ size range. The AIS size distributions indeed show that ions in the 1-10 $nm$ size range appear only one hour after the rise in $H_2SO_4$ concentration. However, the concentration of 10-20 $nm$ particles is not directly linked to the concentration of $H_2SO_4$, although globally $H_2SO_4$ are higher during the May period, as are the concentrations of the 10-20 $nm$ particles. For the case of 20 May, it is also possible that condensable gases concentration is so high that particles are produced and grown to sizes larger than 20 $nm$ already before they reach the Maïdo site. Indeed, large number concentrations

of particles larger than 20 $nm$ are detected on this day. Other factors influence the concentration of particles in the 10-20 $nm$




size range, as the condensational sink (higher during the May period) and coagulation effects. These factors should be taken into account to further calculate nucleation rates and their link to $H_2SO_4$. Then the variability of the correlation between the new particle formation rate and sulphuric acid will be further studied on other plume detection period at the Maïdo, in view of deriving, for the first time to our knowledge, parameterizations adapted for the nucleation process in volcanic plumes.

**9   Conclusions**

The STRAP 2015 campaign was dedicated to the observation and study of diluted volcanic plumes of Piton de la Fournaise volcano. It is probably one of the first experiment campaign during which the measurements permitted to carefully track its downwind evolution from the proximal (near the vent) and to the distal area (over La Réunion island) over tens of $km$. Moreover, plume tracking was performed during two distinct eruptions of Piton de la Fournaise (May 2015 and August-

October 2015) representative of two distinct end-members of the activity of the volcano, the first one being short lived and emitting a low volume of lava and the second one having a long duration, high intensity and a complex time evolution. This campaign, conducted over an extended period, required an important human and scientific investment and a major effort for coordination of a multidisciplinary team. These new exciting results have been made possible by several year-long efforts of cross-disciplinary collaboration between volcanologists and meteorologists, developing shared scientific language, methods

and objectives. After 3.5 years of rest, the reawakening of Piton de la Fournaise in 2014 and the escalation of volcanic activity in 2015 provided unique opportunity of a major collaborative effort among several international and local partners. The purpose of this article was twofold: (i) to present the methodological approach developed to track plume evolution from the source to the distal area, and (ii) to summarize the preliminary observations of gaseous emissions, plume location, height and dispersion and gas-particle conversion. Several new interesting elements already can be identified in these preliminary results.

The DOAS network permitted unprecedentedly detailed and time-resolved measurement of plume geometry (altitude, direction of dispersion) as well as of the rate of $SO_2$ emission during daytime. $SO_2$ fluxes where highest (in the range 5-6 $kt\ day^{-1}$) at the beginning of the largest eruption (25-27 of August) and in the period 25 September to 30 October. Interestingly a peak in $SO_2$ emissions (up to 6.8 $kt\ day^{-1}$) was also recorded at the end of phase 1, in the morning of 10 September (between 3h51 and 5h40 UTC). Preliminary observations on the eruptive products suggest that this peak in $SO_2$ might have heralded a change

in magma composition from phase 1 to phase 2 (Coppola et al., submitted). During most of the eruption, $SO_2$ fluxes have been lower than 1.5-2 $kt\ day^{-1}$.

The derived plume height shows reasonable agreement with simultaneous ultralight aircraft and LIDAR measurements, and some differences with ground meteorological stations, showing meteorological and topographical effects. During the short lived May eruption, the $SO_2$ emissions declined rapidly in time, as often observed in classical eruptions of Piton de la Four-

naise. A much more complex time evolution was traced during the long lasting August-October eruption, showing a marked increase in emission rates in its second half. Preliminary analyses of the erupted lavas highlight the involvement of several magmatic sources during the second large eruptive event, each possibly having a distinct signature in the composition of erupted gases. The MultiGAS measurements of gas emissions were acquired close to the eruptive vent during each eruption, permit-




ting to constrain the time evolution of the ratios between $CO_2$, $H_2O$ and $SO_2$. By correlation with the DOAS measurements, the emission of $CO_2$ and $H_2O$ is therefore established during most of the campaign. Overall, high-temperature water vapour dominates with a molar ratio $H_2O/CO_2$ fluctuating around 50-240 and $CO_2/SO_2$ molar ratio in the range 0.04-0.56. These results are key parameters to be introduced in mesoscale chemical models to represent the life cycle of the volcanic plume.

The LIDAR measurements over the whole year have permitted to draw a climatology of the plume altitude and thickness. The thickness of the plume varies between 700 and 2000 $m$ with a top around 3000 $m$ asl that can reach 4000 $m$ asl during the most intense activity periods. These results are fully consistent with geometrical determination made by the DOAS network. FLEXPART simulations of transport-diffusion of a volcanic tracer allowed drawing up a climatology of the plume distribution on the two most important eruptive periods (May and August-October 2015). Overall, the model shows a marked diurnal evo-

lution. During nighttime and in the early morning, FLEXPART model produces areas with pollutants accumulation in ruggy topography such as the Rempart valley or the Cilaos. In the afternoon the plume is modelled higher in altitude with a more homogeneous distribution due to the increase in thickness of the mixing boundary layer. This diurnal evolution of the volcanic pollutant distribution on the slopes of the volcano far from the volcanic vent was also observed visually in the field (visually sulfur plume appears bluish). The comparisons between the LIDAR measurements and the FLEXPART simulations have en-

abled us to assign some aerosols layers to the volcanic plume. In the distal area, most of the time, the layer associated with the volcanic plume is located at 2000 $m$ asl which roughly corresponds to the altitude of the vent. Compared with DOAS observations, the distal LIDAR measurements generally constrain the altitude of the plume to a lower level than the one measured near the eruptive vent. One explanation could be attributed to a subsidence of the weak eruptive plume by orographic effect during its advection to the northwest. Another reason is due to the definition of plume height, which in the case of DOAS corresponds

to the altitude of the column-weighted centre of mass of the plume and, in case of LIDAR measurements, it corresponds to the base of the plume.

The combination of the sun photometer measurements and of the backscattered signal of the LIDAR gives the aerosol optical thickness, the Ångström coefficient, the LIDAR ratio and the optical extinction profile of the encountered aerosols. In the layers attributed to the freshly emitted volcanic plume, the extinction coefficient reaches 0.20 $km^{-1}$, the Ångström coefficient

between 500 $nm$ and 675 $nm$ is 1.26±0.19 and the LIDAR ratio at 355 $nm$ equals 42±10 $sr$. These last values indicate the dominant presence of small particles and are in agreement with previously published studies. The airborne measurements consistently show the presence of the volcanic plume at about 2000 $m$ asl. $SO_2$ concentrations measures are very high with levels sometimes exceeding 3000 $ppb$ a few $km$ away of the vent. Strong $SO_2$ concentrations above 1000 $ppb$ are frequently measured over 20 $km$ of distance from the Piton de la Fournaise. The measured aerosol size distribution shows the predom-

inance of small particles in areas with high $SO_2$ concentration. The number of particles greater than 7 $nm$ is also very high, reaching 40,000 and 70,000 $cm^{-3}$ when the aircraft flies in the volcanic plume.

Several measurements clearly show the passage of the volcanic plume over the Maïdo observatory. In particular, on 20 and 21 May, the $SO_2$ concentration reached 433 $ppb$. The combined measurement of SMPS, nanoCPC and AIS enabled to highlight a strong rate of sulphuric acid nucleation. The size distribution of the smallest ions (1-10 $nm$) increases in the early

morning (around 6 UTC). The appearance of the cluster ions is rapidly followed by their fast growth from 10 to 50 $nm$ on



1-2 September, and to 100 $nm$ on 20-21 May. An analytic computation of the $H_2SO_4$ concentration derived from radiation and the $SO_2$ concentration indicates that $H_2SO_4$ and the ion concentrations increase simultaneously in the morning. The AIS size distributions show that ions in the 1-10 $nm$ size range appear only one hour after the $H_2SO_4$ concentrations raise. Most important, the preliminary correlation between proximal and distal observation suggests that weak eruptions like May 2015 or

the weaker phases of the long lasting August-October eruption have a stronger impact on the distal site of Maïdo, with respect to the strongest eruptive phases. We speculate that low plume height during weak eruptive phases favours plume subsidence and enhances interaction with the rugged topography of the island, producing a stronger pollution tens of $km$ away from the volcanic source.

Ongoing researches on the unique dataset acquired during the STRAP 2015 campaign will surely help to better understand the

ageing process of plumes emitted by the basaltic volcanoes and their impact on air quality at regional scale. Ash poor gas rich emissions are widespread at volcanoes worldwide both during quiescence and unrest phases. The little amount of silicate ash and primary aerosol in the plumes of Piton de la Fournaise make of this volcano an ideal target to better understand the time and space evolution of volcanic gas. Future researches will focus on the evolution of gas-particles formation and interaction in areas close to the vent with a significant presence of water, and at a greater distance in a drier air mass. The STRAP campaign

is also expected to better constrain the role of the local circulations and their impact on the pollutants accumulation in densely populated areas of the island. These processes will be analysed using a full chemical mesoscale model such as MesoNH used by Durand et al. (2014) for the 2007 eruption of the Piton de La Fournaise. The STRAP campaign will continue on the Etna volcano (Italy) in 2016 with instruments on-board the ATR42 of the French SAFIRE team which will aim at studying the volcanic plume formed by continuous passive degassing. It will then be possible to compare the specificities of these two types

of plume and to test the nucleation and coagulation processes on these two target volcanoes.

*Acknowledgements.* The STRAP project was funded by the Agence Nationale de la Recherche (ANR-14-CE03-0004-04) and the OMNCG/OSU-R program from La Réunion University. The DOAS stations were funded by the FP6 project NOVAC. The authors acknowledge P. Goloub and PHOTONS/AErosol RObotic NETwork (AERONET) for the MICROTOPS II data processing. We also want to thank all the participants involved in STRAP in 2015 from the LACy, the OVPF, the LaMP, the INGV, the UMS 3365 and the ORA. François Levassort is also ac-

knowlegded for his participation to the LIDAR measurements campaign during his vacations. The ultralight measurements would not have been possible without the availability and the kindness of Didier Gouloumès from ALPHA ULM. Warm thanks to Didier.




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





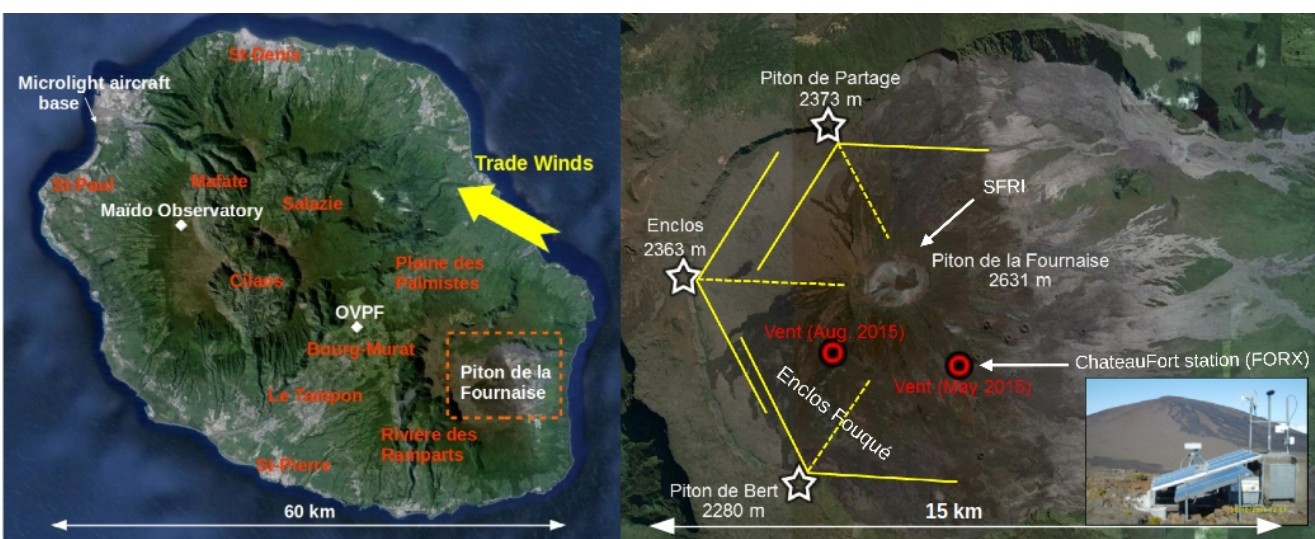

**Figure 1.** Left panel: map of La Réunion island including the sites of interest quoted in the text: the location of the principal cities and of the Piton de La Fournaise volcano (red), and the location of the three major sites of the campaign (white). The yellow arrow indicates the mean direction of the trade winds and the dotted rectangle represents the area shown in the figure at right. Rigth panel: location of the scanning-DOAS stations of the NOVAC/OVPF network (stars; the picture shows the Enclos station). The dashed lines are the azimuths of the main axis for each station and the continuous lines the angular coverage of each scanner. The locations of the main eruptive vents of the May and August-October 2015 eruptions are also shown (concentric dots).





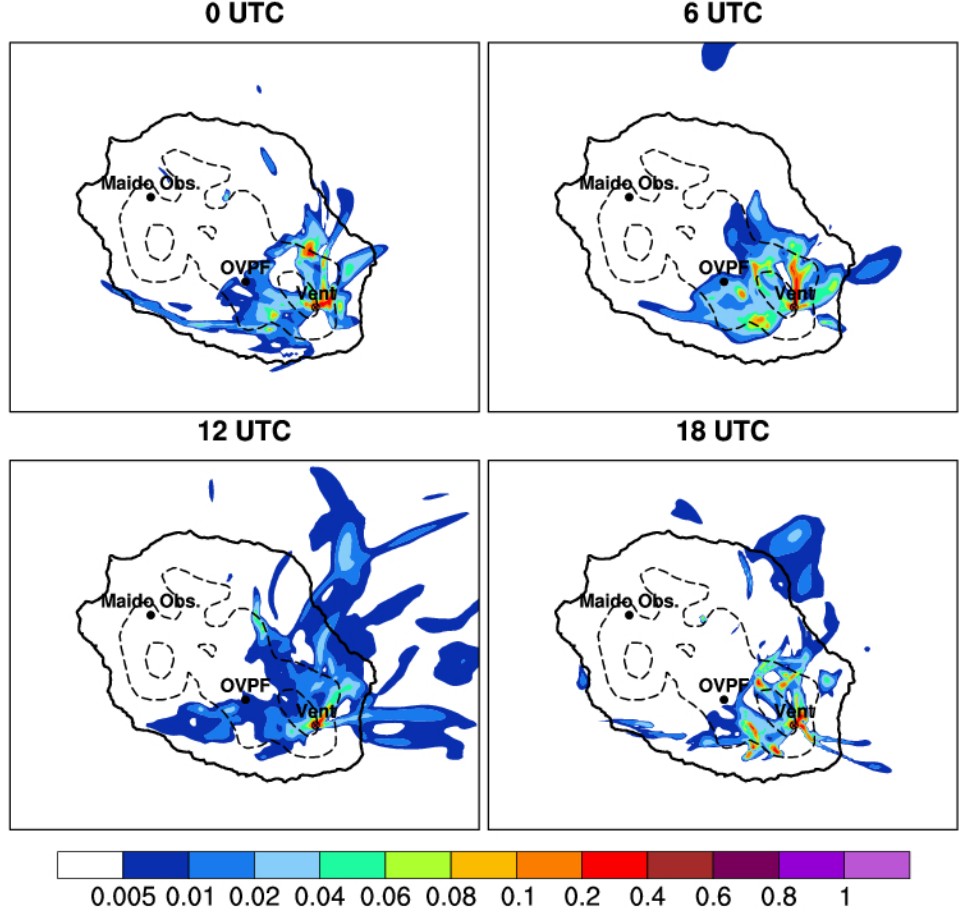

**Figure 2.** Normalized hourly temporal mean of a tracer load integrated over all mode levels (representing the $SO_2$) emitted at the vent and simulated by FLEXPART between 17 May at 00 UTC and 31 May at 23 UTC. The locations of the Maïdo observatory, the OVPF and the vent are pointed on the figure. The dotted lines show the 1000m and 2000m orography level contours.





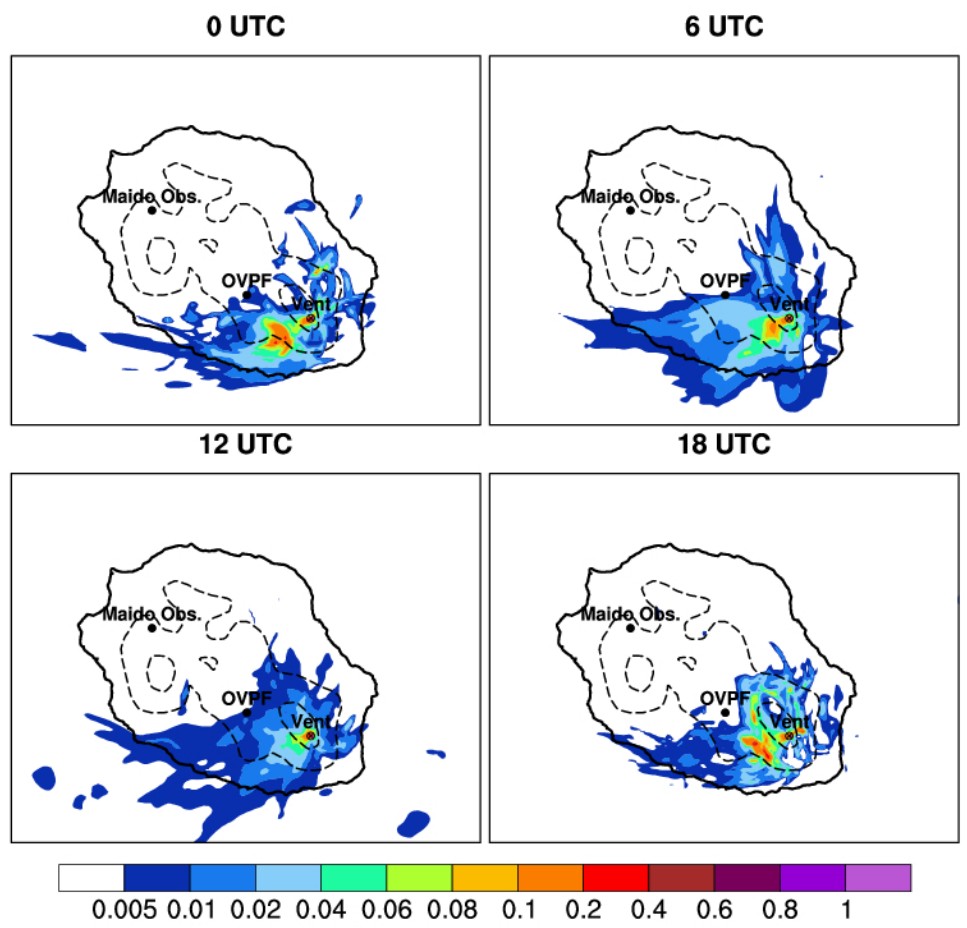

**Figure 3.** Same as 2 but for a simulation between 24 August at 00 UTC and 31 October at 23 UTC.





**Figure 4.** Time evolution of the plume height (top left) and plume direction (top right) computed by triangulation of nearly simultaneous scans of two permanent DOAS stations of OVPF (NOVAC network) and by the Météo-France station. On the bottom left panel; the histogram of plume directions measured by the scanners and wind direction measured by the meteorological station are shown. On the bottom right panel; the difference between plume direction derived by DOAS and wind direction at the Bellecombe Météo-France station against plume height are reported. Wind speed corresponds to the hourly average of the measurements of the Météo-France Bellecombe station.





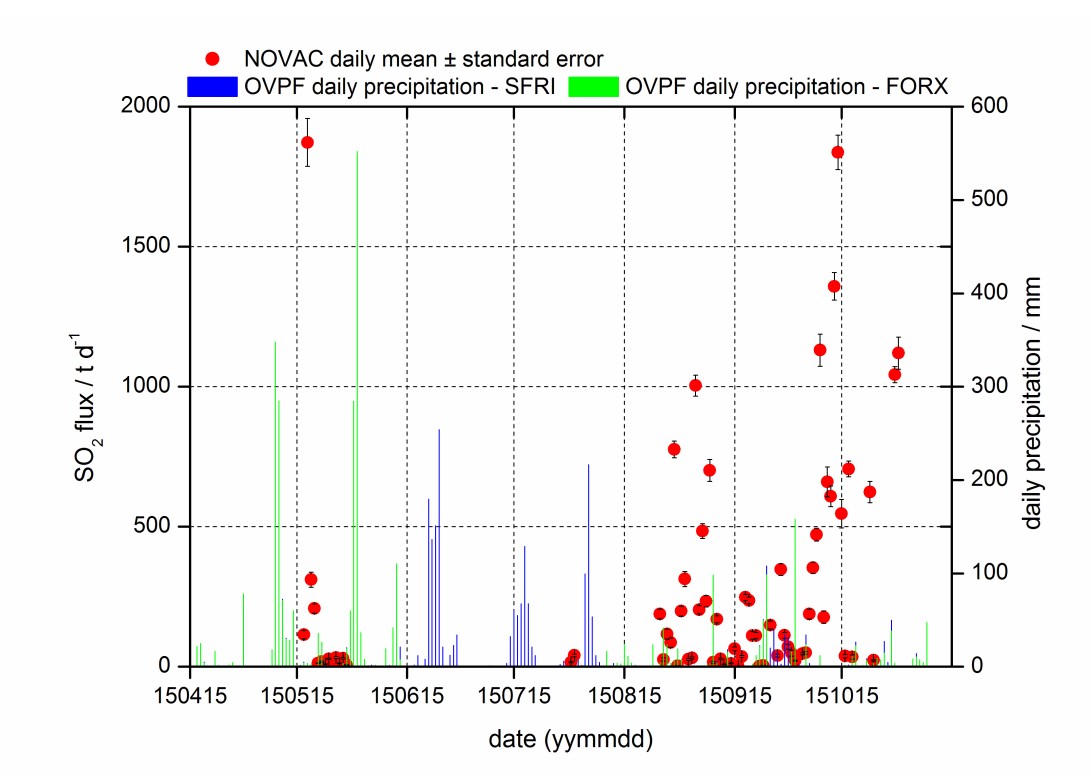

**Figure 5.** Estimated daily mean $SO_2$ flux obtained by integrating the DOAS plume scans with the wind speed provided by Météo-France (Bellecombe station). Daily precipitations are from the SFRI and ChateauFort (FORX) stations of OVPF located respectively at the top and at the SE base of the central cone.





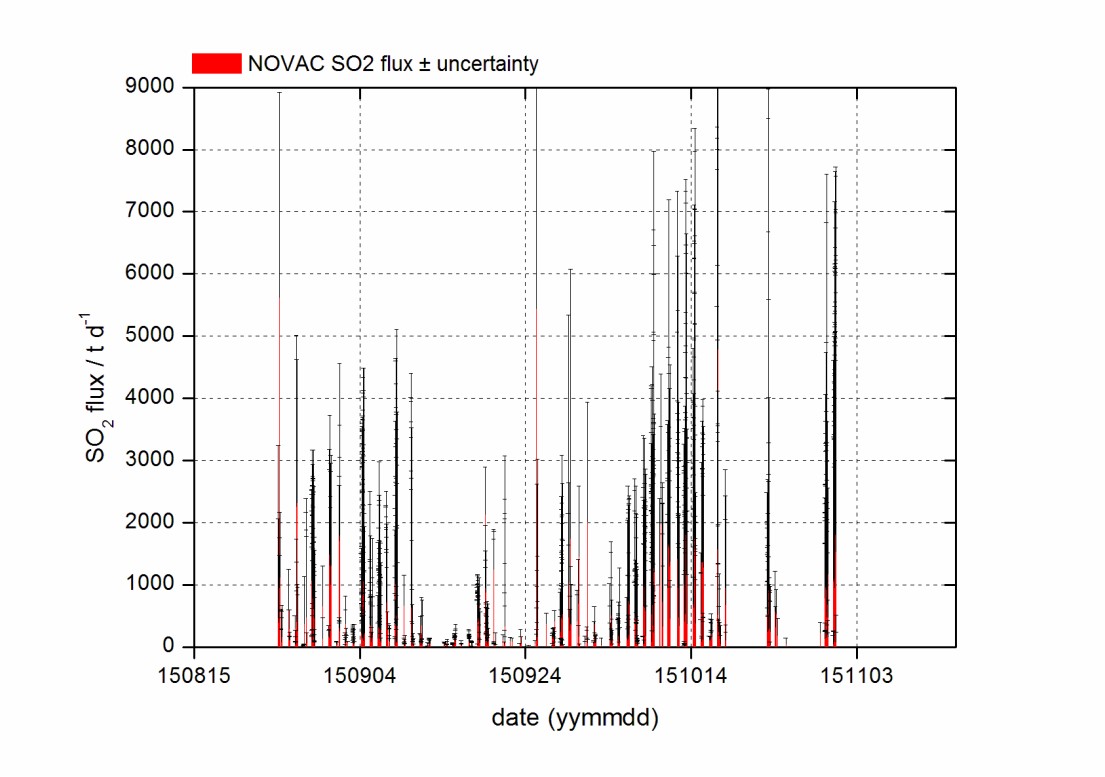

**Figure 6.** $SO_2$ flux from individual scan measurements of the DOAS network (red) during the large August-October 2015 eruption. The uncertainty comes from the spectroscopic retrieval, radiative transfer, wind direction and speed, and plume height. This uncertainty is used in the computation of the daily mean values as presented in Figure 5.

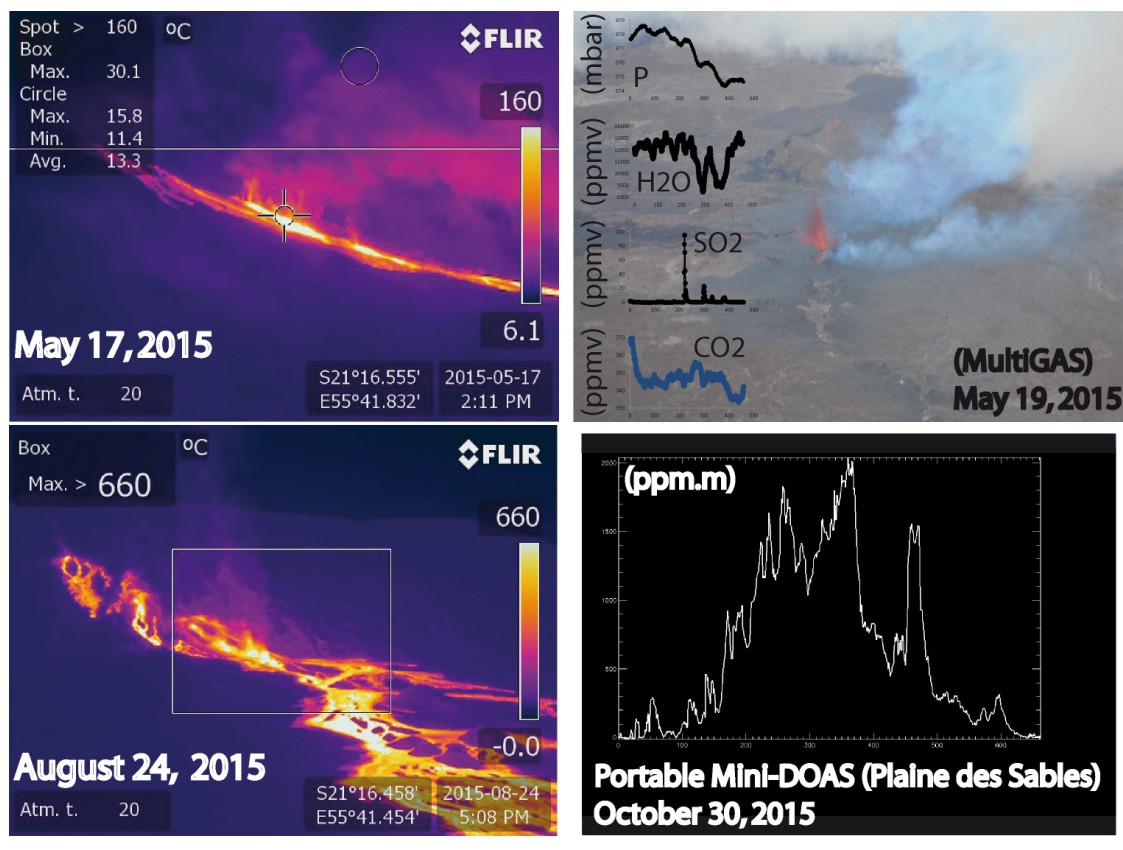

**Figure 7.** Nighttime IR remote imagery (view from the Piton de Bert site on the southern cliff of the Enclos Fouqué caldeira (see Fig. 1) of the linear eruptive sources active at the beginning of May 2015 (top left panel) and August 2015 (bottom left panel) eruptions. Top panel at the right; in situ analysis of volcanic gas emissions by MultiGAS technique carried by an helicopter close to the eruptive vent (May 2015 eruption). Bottom panel at the right; $SO_2$ column density measured by portable mini-DOAS in a traverse of the gas plume (Plaine des Sables site: 4.3 $km$ north-west from the eruptive vent).



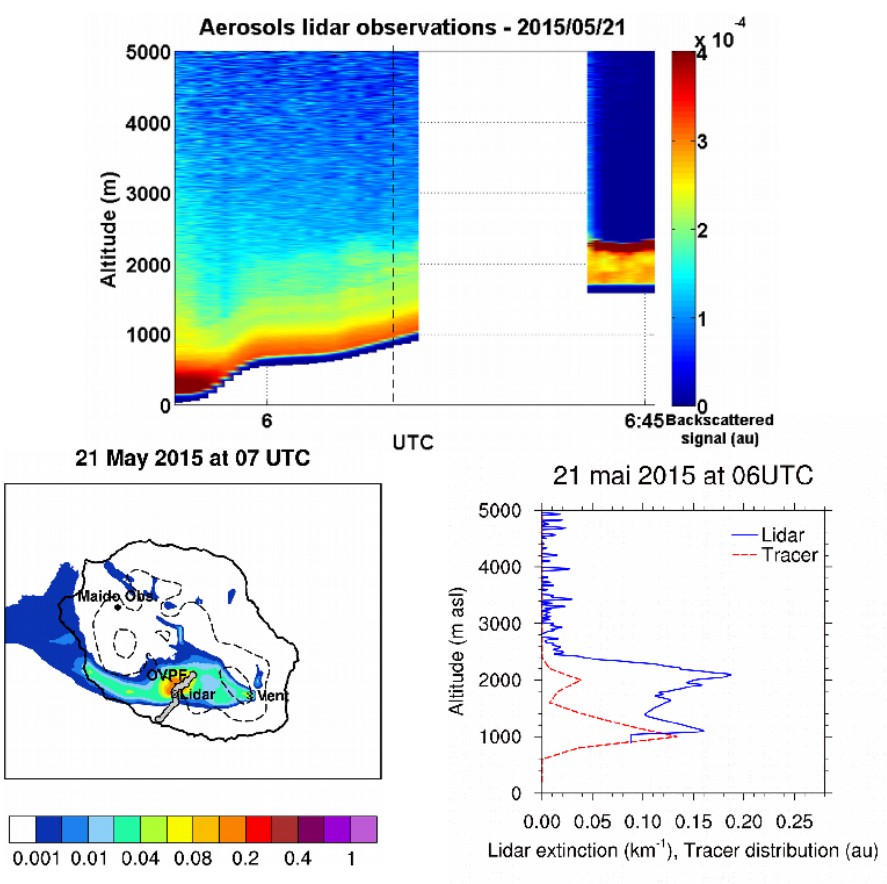

**Figure 8.** Top panel; backscattered signal (arbitrary unit) along the 21 May pick up's track (grey line on bottom left panel). The black dashed line shows the profile used to retrieve the extinction profile given in example in the bottom right panel. Bottom left panel; FLEXPART simulation of the volcanic tracer load modelled on 21 May at 6 UTC and the trace of the pick-up path (grey line). Bottom right panel; aerosols extinction profile retrieved on 21 May 2015 6:14 UTC (blue line) and tracer profile simulated by FLEXPART at the same location at 6:00 UTC (dashed red line).





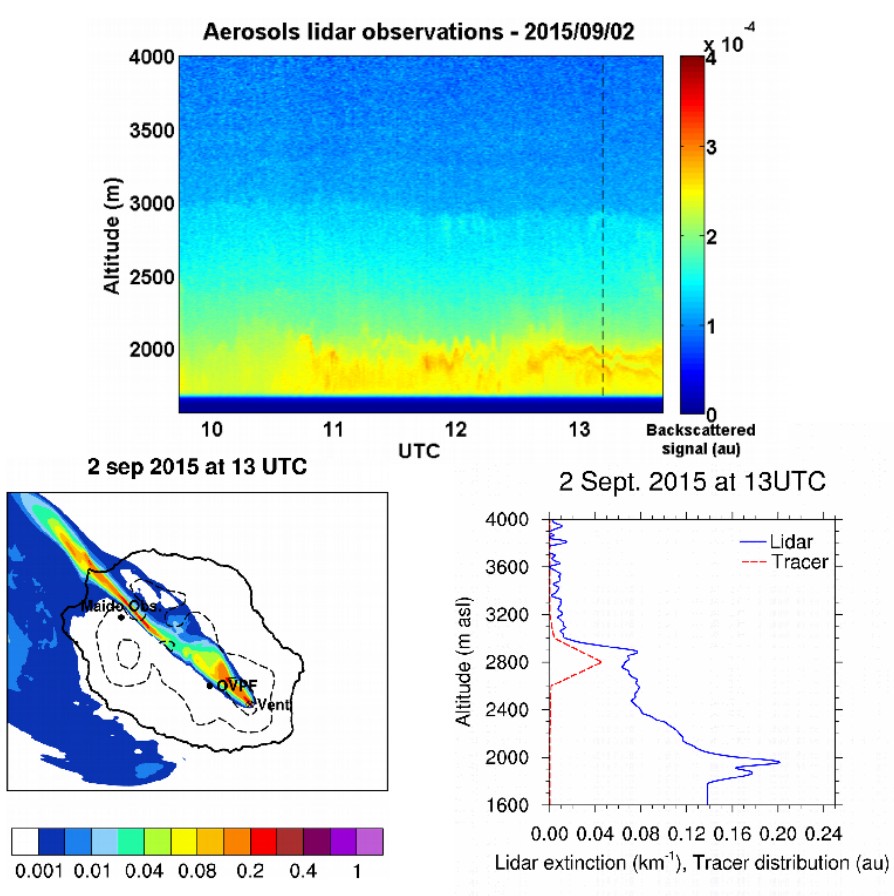

**Figure 9.** Top panel; time serie of the backscattered signal (arbitrary unit) and profile (black dashed line) on 2 September 2015 used to retrieve the extinction profile given in example in the bottom right panel. Bottom left panel; FLEXPART simulation of the volcanic tracer load modelled on 2 September 2015 at 13 UTC and location of the pick-up (OVPF). Bottom right panel; aerosols extinction profile retrieved on the 2 September 2015 13:10 UTC (blue line) and tracer profile simulated by FLEXPART at the same location at 13 UTC (dashed red line).




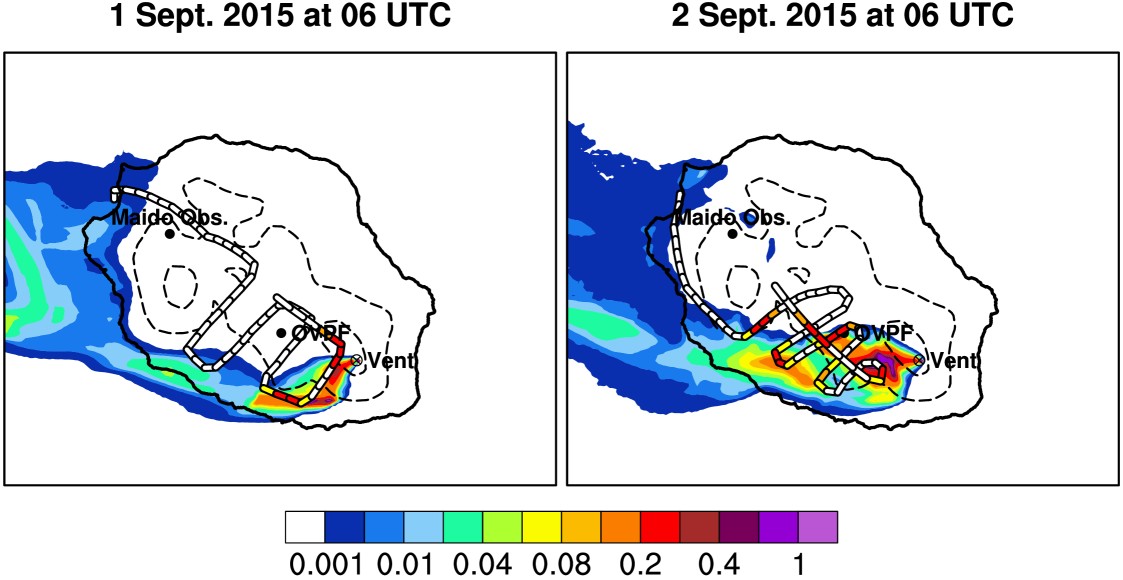

**Figure 10.** Concentration of the tracer load emitted at the vent and simulated by FLEXPART on 1 September 2015 (left) and on 2 September 2015 (right). The contour lines in colour represent the trajectory of the aircraft flight in function of the measured $SO_2$ concentration (white for less than 100 $ppb$, yellow between 100 and 200 $ppb$, orange between 200 and 400 $ppb$ and red for more than 400 $ppb$).





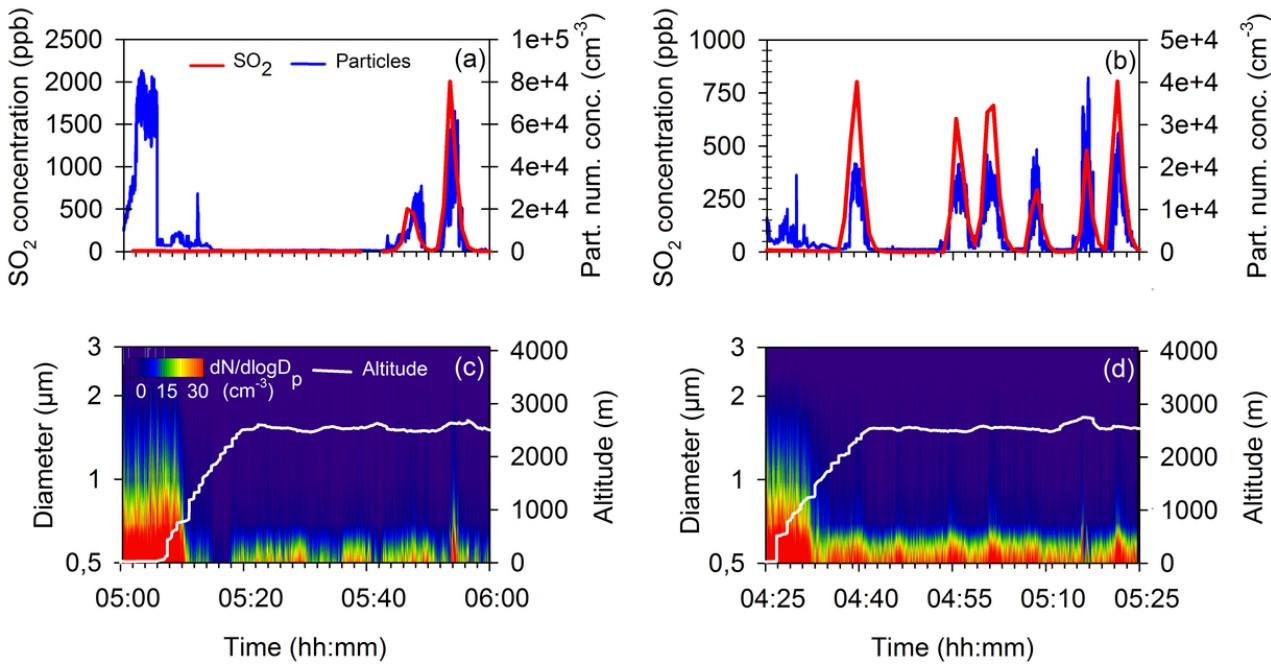

**Figure 11.** Onboard microlight aircraft measurements sampled on 1 September 2015 (left panel) and 2 September 2015 (right panel) over La Réunion. At the top; time series of $SO_2$ concentration (isoline in red) and particle number concentration (isoline in blue). At the bottom; particle number size distribution (colour range) and altitude of the aircraft (in white). The trajectories of the flights are pointed out on Figure 10.





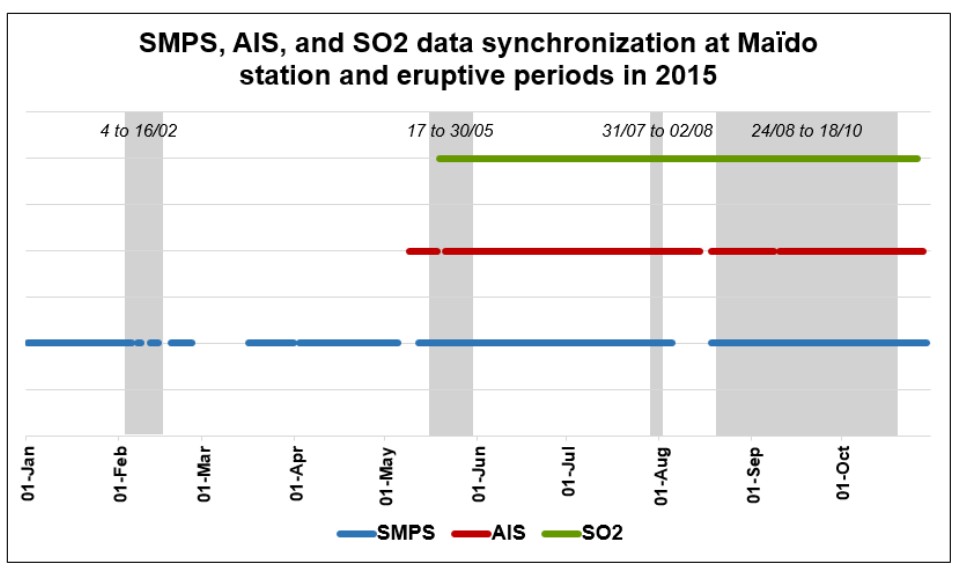

**Figure 12.** Data coverage of instrumentation for the characterization of new particle formation events during the eruption period (in grey) of May to October 2015.

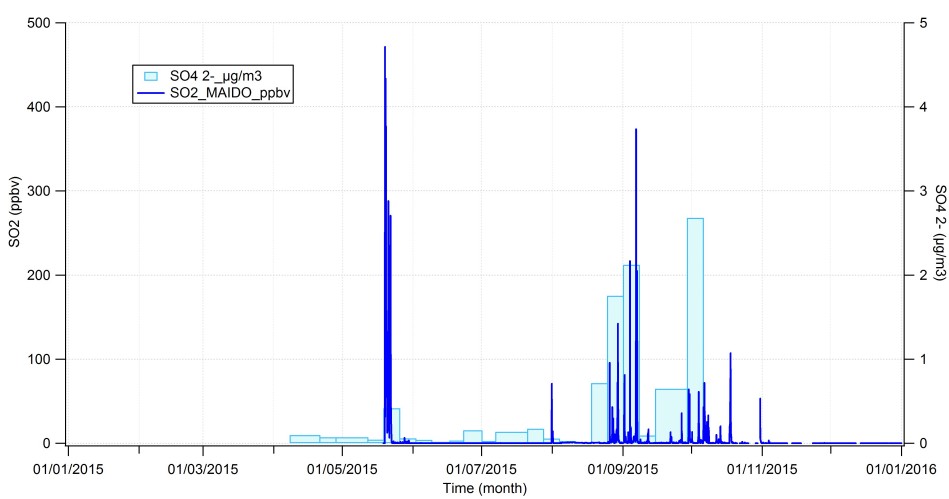

**Figure 13.** $SO_2$ concentration time series measured from the Maïdo observatory during the year 2015. Measurements started on 19 May. $SO_4^{2-}$ from filter sampled at the Maido observatory integrated each week during night (between 18 and 1 UTC).



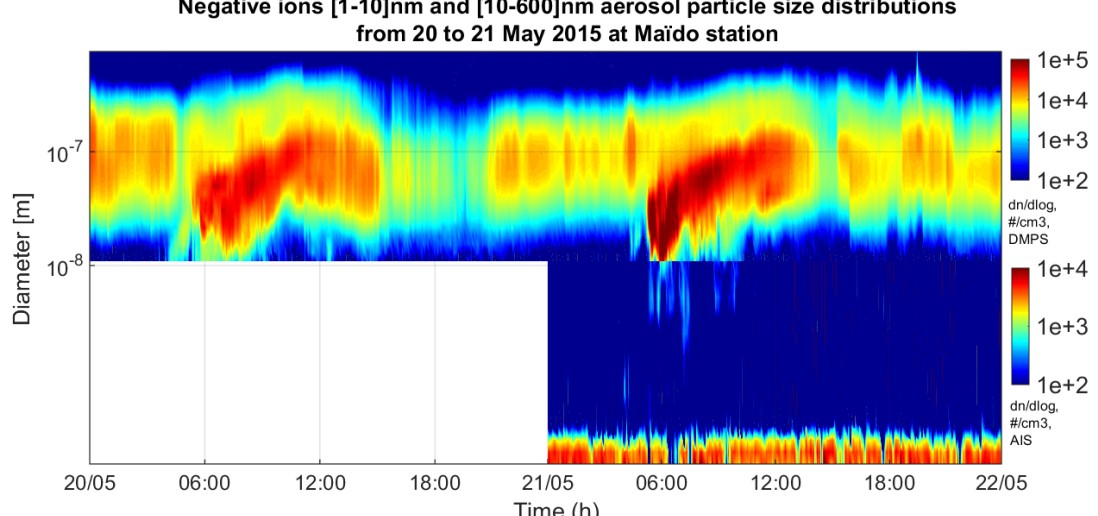

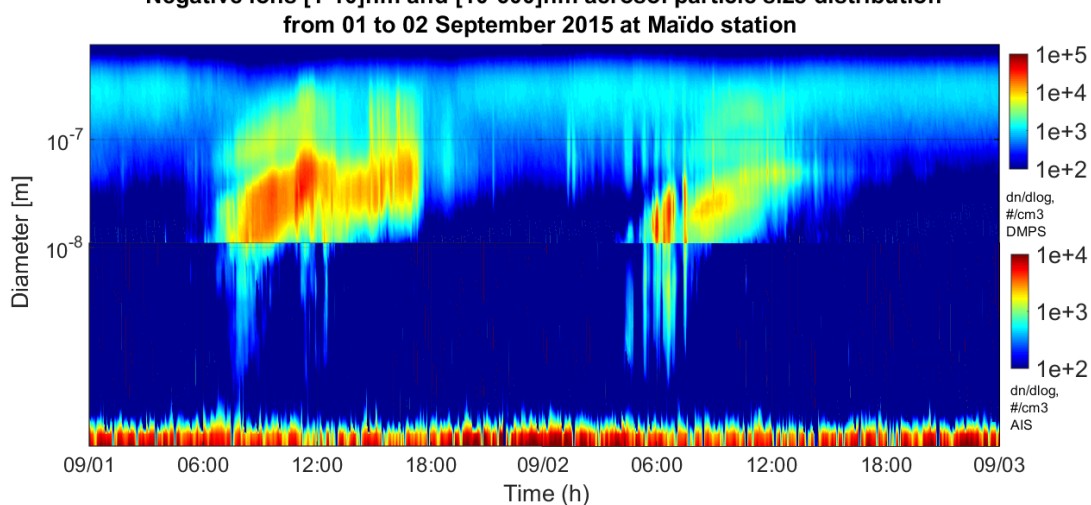

**Figure 14.** Negative ions (1-10 nm) and (10-600 nm) aerosol particle size distributions measured (top panel) on 20-21 May 2015 and (bottom panel) on 2-3 September when the Piton de la Fournaise plume crossed the Maïdo station. No AIS data between 20 May and 21 May at 5 UTC. Color scale (right) indicates the ratio dN/dlog in $cm^{-3}$



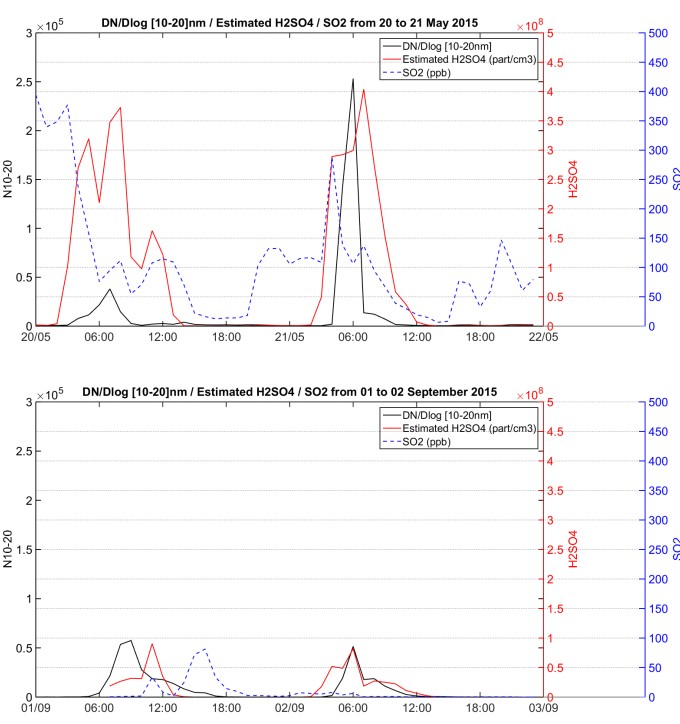

**Figure 15.** Time series of the 10-20 $nm$ particle concentrations (black), $SO_2$ concentrations (blue) and $H_2SO_4$ concentrations estimated by proxy (red) on 20-21 May 2015 (top) and on 1-2 September 2015 (bottom).





**Table 1.** Summary of in-situ chemical measurements (Multigas, Portable Mini-DOAS and FLIR camera) made on field near the vent on May-October 2015. The symbol X indicates when measurements using FLIR camera have been done.

| Date (MM/DD/YY) | MultiGAS local time; coordinates | Portable Mini-DOAS local time | Portable FLIR camera |
|---|---|---|---|
| 5/17/2015 | 16-16:24; (close to lava field; -21°15'58.9", 55°42'59.8")) | - | X |
| 5/18/2015 | 13:45-14:06; (near eruptive vent; -21°15'26.0", 55°43'28.4") | 12:01 | - |
| 5/19/2015 | 13:26-13:41; helicopter near eruptive vent | 13:25 | X |
| 5/20/2015 | 15:15-15:41; near eruptive vent | - | X |
| 5/21/2015 | | 10:26 | - |
| 5/24/2015 | 12:22-12:50; near eruptive vent | - | - |
| 8/24/2015 | | - | X |
| 8/25/2015 | 11:42-12:12; (-21°15'19.1", 55°42'14.9") | - | - |
| 8/27/2015 | | - | X |
| 8/27/2015 | 12:22-12:40; (-21°15'19.1", 55°42'14.9") | - | - |
| 8/27/2015 | 10:25-10:45; (close to the vent exit and along the lava channel: -21°15'24.2", 55°42'15.9" to -21°15'31.2", 55°42'11.7" | - | - |
| 8/28/2015 | 14:20 (Plaine des Sables) | - | - |
| 9/01/2015 | | - | X |
| 9/03/2015 | | - | X |
| 9/03/2015 | 11:45-12:45; close to active eruptive fissure | - | - |
| 09/07/2015 | 12:15-13:00; close to active eruptive fissure | 12:22 | - |
| 09/08/2015 | | - | X |
| 09/11/2015 | | - | X |
| 09/11/2015 | 11-12:36; close to the only active cone | 11:17 | - |
| 09/15/2015 | 16:30-17:05; close to the only active cone | - | X |
| 09/18/2015 | 12-13:24; close to the only active cone | 12:22 | X |
| 10/07/2015 | 11:00-11:30; close to the only active cone | - | - |



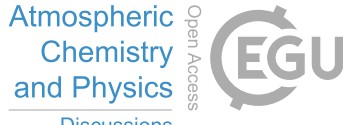

| | | | |
|---|---|---|---|
| 10/09/2015 | | - | X |
| 10/11/2015 | | - | X |
| 10/16/2015 | | - | X |
| 10/22/2015 | | - | X |
| 10/23/2015 | 12:40-13:34; close to the only active cone | - | X |
| 10/26/2015 | 08:30-09:04; on the rim of the active crater during the rest phase (26/10) | - | X |
| 10/30/2015 | 10:37; 11:27 (Plaine des Sables) | - | - |
| 10/31/2015 | 15:37-15:56; close to the feeding outbreak of the new lava outburst | - | - |



**Table 2.** Summary of flights made by the microlight aircraft during the volcanic ash events of May-September 2015.

| Date | Take off time (UTC) | Acquisition duration (mn) | Maximum $SO_2$ concentration (ppb) | Maximum number concentration of particles $(cm^{-3})$ | Comments on data collected |
|---|---|---|---|---|---|
| 19/05/2015 | 04:47 | 60 | - | 44680 | No $SO_2$ data |
| 20/05/2015 | 04:53 | 60 | 744.5 | 33554 | Good sampling |
| 22/05/2015 | 04:54 | 55 | 382.8 | 19837 | Good sampling |
| 25/05/2015 | 04:39 | 60 | 282.8 | 18500 | Good sampling |
| 26/05/2015 | 03:57 | 51 | 33.7 | 436 | Good Sampling, end of the eruption |
| 01/08/2015 | 05:38 | 60 | 2530.0 | 60310 | Good sampling |
| 25/08/2015 | 07:21 | 60 | - | 48700 | No $SO_2$ data |
| 26/08/2015 | 07:12 | 60 | - | 31020 | No $SO_2$ data |
| 27/08/2015 | 06:11 | 63 | - | 73020 | No $SO_2$ data |
| 28/08/2015 | 04:57 | 60 | 105.7 | 14510 | Good sampling |
| 29/08/2015 | 03:34 | 64 | 3330.0 | 65850 | Good sampling |
| 31/08/2015 | 04:42 | 60 | 669.0 | 31710 | Good sampling |
| 01/09/2015 | 04:56 | 58 | 2004.0 | 65250 | Good sampling |
| 02/09/2015 | 04:25 | 60 | 804.5 | 43450 | Good sampling |
| 03/09/2015 | 04:29 | 60 | 626.3 | 41270 | Good sampling |
| 10/09/2015 | 07:19 | 67 | 2339.0 | 17367 | Good sampling |
| 11/09/2015 | 04:39 | 78 | 12.7 | 5025 | Good sampling |
| 18/09/2015 | 05:02 | 66 | 472.0 | 43130 | Good sampling |





**Table A1.** Summary of material collected on field near the vent on May-October 2015. Contact the OVPF/IPGP for further details about the samples.

| Date (MM/DD/YY) | Material type | Location | Weight (gr) | Note on Texture/Size |
|---|---|---|---|---|
| 5/17/2015 | lava | -21°15'58.9", 55°42'59.8" | 1210.7 | lava |
| 5/18/2015 | lava | -21°15'26.0", 55°43'28.4" | 695.7 | pyroclasts of the fountain |
| 5/20/2015 | lava | -21°15'36.36", 55°43'37.93" | 311.7 | lava |
| 5/20/2015 | lava | -21°15'32.21", 55°43'27.45" | 312.0 | pyroclasts of the fountain |
| 5/20/2015 | lava | -21°15'32.21", 55°43'27.45" | 313.2 | pyroclasts of the fountain |
| 5/24/2015 | lava | -21°15'28,75", 55°43'32,62" | 1280 | lava |
| 5/24/2015 | lava | 3674277649085 | 214 | scoria |
| 5/24/2015 | lava | 200 $m$ North-West of Chateau-Fort | 289 | lava |
| 5/24/2015 | fumaroles | 3674277649085 | 264 | fumarolic deposit |
| 5/27/2015 | lava |  | 682 | lava |
| 8/25/2015 | pyroclast | -21°15'19.1", 55°42'14.9" | 296.0 | pyroclast of the intense cone (middle) |
| 8/25/2015 | pyroclast | -21°15'17.9", 55°42'16" | 201.0 | pyroclasts of the strombolian cone (upper) |
| 8/27/2015 | lava | -21°15'31.6", 55°42'11.4" | 213.0 | lava |
| 8/27/2015 | pyroclast | -21°15'31.6", 55°42'11.4" | 737.0 | bombs |
| 8/27/2015 | lava | 3654557648678 | 696.0 | lava |
| 8/27/2015 | pyroclast | -21°15'19.1", 55°42'14.9" | 86.0 | pyroclast of the intense cone (middle) |
| 8/27/2015 | pyroclast | -21°15'18.6", 55°42'12.3" | - | lapilli |
| 8/27/2015 | pyroclast | -21°15'18.6", 55°42'12.3" | - | lapilli |
| 8/27/2015 | pyroclast | -21°15'19.1", 55°42'14.9" | 678 | lapilli |
| 8/27/2015 | fumaroles | -21°15'13.9", 55°42'15.5" | 428 | on lava |



| | | | | |
|---|---|---|---|---|
| 8/28/2015 | pyroclast | -21°14'25.2", 55°42'16.2" | <20 | ash + lapilli |
| 8/28/2015 | pyroclast | -21°14'26.0", 55°42'13.1" | 426 | ash + lapilli + bombs |
| 8/28/2015 | pyroclast | -21°14'26.0", 55°42'13.1" | <10 | lapillis |
| 8/28/2015 | pyroclast | -21°15'00.3", 55°42'18.5" | <15 | lapillis |
| 8/28/2015 | pyroclast | -21°14'59.9", 55°42'17.6" | 799 | lapillis + bombs |
| 8/28/2015 | pyroclast | -21°15'19.1", 55°42'14.9" | 330 | lapillis |
| 9/01/2015 | lava | -21°15'29.28", 55°42'14.46" | 1236 | lava |
| 9/03/2015 | lava | -21°15'26.1", 55°42'12.9" | 713 | lava |
| 9/03/2015 | pyroclast | -21°15'19.1", 55°42'14.9" | 298 | lapilli + bombs (dense + golden pumice) |
| 09/07/2015 | pyroclast | -21°15'18.1", 55°42'11.7" | 625 | lapilli + bombs (dense) |
| 09/07/2015 | pyroclast | -21°15'18.1", 55°42'11.7" | 89 | lapilli + bombs (golden) |
| 09/07/2015 | pyroclast | -21°15'18.1", 55°42'11.7" | <27 | Pele's hairs + fines + lapilli |
| 09/07/2015 | pyroclast | -21°15'18.1", 55°42'11.7" | 433 | lapilli + bombs |
| 09/11/2015 | pyroclast | -21°15'18.1", 55°42'11.7" | 1013 | bombs |
| 09/11/2015 | pyroclast | -21°15'18.1", 55°42'11.7" | 88 | lapillis + ash |
| 09/15/2015 | pyroclast | -21°15'18.1", 55°42'11.7" | 338 | lapillis + bombs |
| 09/18/2015 | lava | -21°15'24.8", 55°42'15.5" | 1756 | partially molten lava crust |
| 09/27/2015 | lava | -21°15'33.1", 55°42'03.35" | 853 | lava |
| 10/07/2015 | lava | -21°15'22.6", 55°42'11.9" | 1280 | lava |
| 10/07/2015 | pyroclast | -21°15'18.1", 55°42'11.7" | <17 | Pele's hairs + fines + lapilli |
| 10/07/2015 | lava | -21°15'22.6", 55°42'11.9" | 1338 | lava |
| 10/09/2015 | lava | 3654407647631 | 829 | lava |
| 10/09/2015 | lava | 3654327647655 | 653 | lava |





| | | | | |
|---|---|---|---|---|
| 10/16/2015 | lava | -21°15'18.1", 55°42'11.7" | 808 | lava (spatter) |
| 10/16/2015 | lava | -21°15'18.1", 55°42'11.7" | 335 | lava (spatter) |
| 10/23/2015 | lava | -21°15'22.6", 55°42'11.9" | 440 | lava |
| 10/23/2015 | lava | -21°15'18.1", 55°42'11.7" | 66 | lapillis |
| 10/23/2015 | lava | -21°15'18.1", 55°42'11.7" | 80 | ash+lapillis |
| 10/26/2015 | pyroclast | 365611E7649014 | 1044 | lapillis |
| 10/26/2015 | pyroclast | 365611E7649014 | 580 | bombs, lapillis |
| 10/26/2015 | pyroclast | 365611E7649014 | 472 | bombs |
| 10/26/2015 | pyroclast | 365476E7649004 | 1670 | bombs |
| 10/26/2015 | lava | 365635E7648935 | 580 | pahoehoe upper crust; tip of the flow |
| 10/26/2015 | pyroclast | 365635E7648935 | 250 | lapillis |
| 10/26/2015 | pyroclast | 365635E7648935 | 662 | bombs, lapillis |
| 10/31/2015 | lava | -21°15'22.6", 55°42'11.9" | 440 | lava |
| 10/31/2015 | lava | -21°15'22.6", 55°42'11.9" | 440 | lava |
| 10/31/2015 | lava | -21°15'22.6", 55°42'11.9" | 440 | lava |
| 10/31/2015 | lava | -21°15'22.6", 55°42'11.9" | 440 | lava |