# Peer review of "First results of the Piton de la Fournaise STRAP 2015 experiment: multidisciplinary tracking of a volcanic gas and aerosol plume"

_Atmospheric Chemistry and Physics, 2016_

## Referee Comment (RC1) · Anonymous Referee #1 · 13 Dec 2016

First results of the Piton de la Fournaise STRAP 2015 experiment: multidisciplinary tracking of a volcanic gas and aerosol plume.

P. Tulet, A. Di Muro, A. Colomb, et al.

This paper present a multidisciplinary approach aiming to track changes during volcanic plume dispersal downwind. The approach combines gas flux and composition measurements, aerosols assessment and volcanic plume dispersal modeling. This combined investigations covered four eruptive periods of Piton de la Fournaise and emphasized existent of a stronger interaction of weak eruptive plume with the island surface, high SO2 and particle concentrations are regularly measured to a distance of 20 km from the source. The study further indicates a predominance of a rather small

particles in the plume in relation to a strong nucleation of sulfuric acid at around 40 km from the source downwind.

This multidisciplinary approach is a step forward to better constrain the physical and geochemical interactions between volcanic plumes and the troposphere. I would criticize the gas composition and fluxes results which appear to me as the weakest part of this work. But the paper is well written and methods are robust and complementary, thus suitable for publication in ACP after considering the few questions and comments below.

1/ Main remarks

This paper strongly emphasized the gas composition and flux measurements but results presented somewhat fail to fulfill expectations:

a) portable DOAS measurement is reported in the paper (L31 p4, L19 p7, L6 p9) but no corresponding result presented. Why? Figure 7 even present a transect across the plume. What is the corresponding SO2 flux? Table 1 indicates 6 series of DOAS measurements but why no preliminary results presented ?

b) August 2015 eruption is described as following 3 different phases, based on results from DOAS stations (L2-6 p10). But the uncertainties associated to these results are significant (Fig.6). - The eruption phase 1 described as associated to a progressive SO2 flux decreasing trend from 24/08 to 12/09 (L4 p10) is not convincing - this tendency is not clearly decreasing (Fig.6). These gas flux results (Fig.5 and Fig.5) will gain more strength if portable DOAS results are associated. - An "accelerating increase of SO2 flux between 13/09 and 18/10" is somewhat exaggerating. According to Fig.6 and accelerating tendency rather commenced in early October. Figure 6 indicates at least two strong degassing phases: from end august to mid-september and from early October to mid-October. Vigorous intermittent SO2 discharges were recorded between and after these two strong degassing phases.

c) MultiGAS measurements is outline several time in the paper (L32 p8, L33 p10, L1 p11, L6 p11, L13 p11, L31 p17,...) and table indicates a total of around 8h of recording from May to October 2015. But curiously only 2 ratios are provided : $H_2O/CO_2$ = 50-240 (L12 p11) and $CO_2/SO_2$ <0.6 (L13 p 11). - It is well known that $H_2O$ and even $CO_2$ are not easily measured in the plume. What is the error of this ratios ? A figure of the plots should be very informative. - Figure 7 gives concentration results which are not exploitable. The behavior of $H_2O$, $CO_2$ and $SO_2$ are totally different which may suggest no common source, that is surprising given that some of the measurement are performed close to the vent. - Should we understand that $H_2O/CO_2$ and $CO_2/SO_2$ ratios are unchanged over the eruptive period ? That would be very surprising given the dynamic of the eruptive activity. Authors should add more results of multiGAS measurements and check the ratio changes which might describe better the eruption dynamic than the $SO_2$ flux from the stationary DOAS. - L32 p 4 indicates $H_2S$ was also measured. But curiously no result mentioned this gas. Is this suggest no $H_2S$ in the system ? That would be very surprising.

2/ Minor remarks - L25 p4: accumulation chamber for $CO_2$ soil flux. Is this instrument deployed ? Not referring to in the rest of the paper. Add reference if developed elsewhere. - L21 p6, delete 2 after August. - L21 p6, the date format is e.g., 2 August 2015 whilst L22 p6 the format is e.g., August 24, 2015. Harmonize date format throughout the paper. L21 p6, to the south-southeast ?....to the north ? L22 p6, to the west-southwest? What are these direction referred to ? L31-32 the output budget ? Not calculated in the paper, why? Do add reference if done elsewhere. L12 p9, DOAS sessions are acquired with a high rate – what does it mean by high rate ? L25-27 p9, 1870 t/d and 1840 t/d is that same if taking into account the errors. Thus not so sure that highest $SO_2$ emission rate was observed on 20 May – maybe tone down this comparison. L31 p9, May $SO_2$ fluxes are not in fig.6, but fig.5 – do modify the sentence. L35 p9, add reference to the estimated 24-37 m3/s, or give further details if calculated in this work.

L4-5 p10 these phases are not convincing – modify to be coherent at least with Fig.6 L12 p10, change (<300 and 600 ton day-1, respectively) to (<300-600 ton day-1) L31 p10 change regular survey and visible and thermal imagery to visible and thermal imagery surveys... ? L32 mentions permitted to follow the time and space evolution of eruptive dynamics and to constrain the evolution of the degassing source – What are the thermal observation results? what can be said about the eruptive dynamics ? What about the evolution of the degassing source ?

L20 p14, change tranformations to transformations

Figure 7. The multiGAS panel is not contributing anything. Worse it presents incomplete (no H2S) and bad (no correlation) results. Add the SO2 flux on the DOAS panel. The changes in thermal images is not described at all, even in the text? Can you say anything ?

---

## Referee Comment (RC2) · Anonymous Referee #2 · 10 Jan 2017

This article presents results from multi-disciplinary observation of volcanic plumes during five eruptive episodes in 2014-2015. There is no doubt that this is an excellent dataset with the potential to aid our understanding of plume development. In particular, the potential of the distant (Maido station) measurements for parameterising nucleation in the plume is exciting.

My main comment about this study is that I found it hard to follow – the introduction to the paper is verbose and discusses the motivation for the whole STRAP project rather than that specifically relevant to the results presented here. The results are separated out into separate sections that also describe methodological considerations for each method. Synthesis and comparison of the different results is not introduced until the

conclusions, and is then very brief.

On page 17, line 16 (conclusions) the authors write that "The purpose of this article was twofold: (i) to present the methodological approach developed to track plume evolution from source to the distal area, and (ii) to summarize the preliminary observations of gaseous emissions, plume location, height and dispersion and gas-particle conversion" This would be a much clearer structure than the outline followed from page 3 lines 19-32 and currently followed. My suggestion is that the article is restructured on the following basis:

1) The introduction could be shortened and made more relevant to the results presented here, e.g., on page 3 lines 9 -19 are dominated by affiliations of the co-authors and some other information that could be in the acknowledgements; section 3.1 is long and could be condensed into an introduction to the more relevant material in section 3.2.

2) The methods of the STRAP experiment, for which results are presented in this article, should be outlined in a methods sections. This should include a clear account of the temporal coverage of the observations that are being presented here (e.g., in a table or figure) It would help the article's readability if methods were clearly linked to the stated goal of tracking plume evolution from source to distal area.

3) The results should be described in a separate section and could be subdivided into (1) a presentation of preliminary observations of plume properties with clear reference to figures and (2) a synthesis of measurements relevant to understanding plume evolution. At present some of the results are well represented by figures, while others are described but not shown.

4) Discussion and conclusions should place the new observations made from measurements presented in this paper into context of past studies at Piton de la Fournaise and other volcanoes.

Line by line comments:

Abstract: line 5 – do measurements span 85 days in total? Does this include gaps in activity? Line 11 – 'a particular emphasis is placed on...' this is an ambiguous phrase. Do you mean that this is a particularly interesting result? How do the SO2, CO2 & H2O levels compare to past measurements/periods of activity. What are the implications for plume interaction with the atmosphere? Are there implications for understanding the development of the eruption (e.g., from increase in SO2 at end of phase referred to later?)

I understand from Section 3.2 that observations from 20th June 2014 to October 2015 are presented in the paper, but from the Figures (Especially 12 and 13) it looks like data were only acquired in 2015 ( the abstract refers to 85 days of measurements and from page 3 line 28 (and figures) it sounds like only the climatology for two eruptions is described). Overall I found it difficult to get my head around the differences in temporal coverage of all the different measurement types – I suggest that the authors include a table, or perhaps a figure, to compare the duration and temporal coverage of each type of observation.

Page 2 line 7-8. Use of 'on one side', 'on the other side' is confusing – these are not opposing ideas? Page 2 line 18. Please add reference for impossibility of obtaining source parameters at Eyja? Page 2 line 31: 'an objective' - is this the particular aim of this work? The following sentence refers to real-time measurement, which I think is not necessary for these goals. page 3, line 5: I don't think this is true. The Boulon 2011 paper does not include measurements made within a volcanic plume. And there are certainly other earlier studies that present measurements of aerosol within volcanic plumes (e.g., Mather et al., 2004; Rose et al., 2006; Martin et al., 2008). Although it provides no information about nucleation mechanism Ebmeier et al., 2014 also shows that there is elevated aerosol and depressed cloud droplet size downwind of PdlF in satellite retrievals averaged over a decade (and a greater effect for periods of eruption). It would be interesting to know how these course observations compare to your much
Interactive
comment

more detailed multi-sensor measurements. Page 4, line 2: 'unique and craggy' is uninformative, 'benefits from a tropical climate softened by the breezes of the Indian Ocean' is also rather informal in style. Section 2.2 title: 'means'=methods? Page 4 line 19: what kind of imagery? Photographs? Figure 1: Resolution appears to be quite low for the size of Figure. Caption: Page 5 line 20 range of dates is surprisingly precise page 10 line 10: I'm not sure that this is an acceleration? Page 17 line 21 – where are these geometries shown? Page 18 line 10 → rugged? Conclusions: Comparison to the previous level of knowledge about the PdlF plume would be useful here - both to place your results in context and help the reader appreciate the level of advance in knowledge offered by such an integrated multi-methodological approach.

Through the article there are English phrases that are ambiguous and some rather awkward constructions. I suggest that English language proof reading would help the final version of this article.

References:

Martin, R. S., et al. "Composition resolved size distributions of volcanic aerosols in the Mt. Etna plumes." Journal of Geophysical Research: Atmospheres 113.D17 (2008).

Mather, T. A., et al. "Characterization and evolution of tropospheric plumes from Lascar and Villarrica volcanoes, Chile." Journal of Geophysical Research: Atmospheres 109.D21 (2004).

Rose, William I., et al. "Atmospheric chemistry of a 33–34 hour old volcanic cloud from Hekla Volcano (Iceland): Insights from direct sampling and the application of chemical box modeling." Journal of Geophysical Research: Atmospheres 111.D20 (2006).

Ebmeier, S. K., et al. "Systematic satellite observations of the impact of aerosols from passive volcanic degassing on local cloud properties." Atmospheric Chemistry and Physics 14.19 (2014): 10601-10618.

---

## Referee Comment (RC3) · Anonymous Referee #3 · 12 Jan 2017

This interesting paper represents a large amount of coordinated work with the aim of tracking plume evolution from the vent to downwind areas. It reports observed plume characteristics as measured by multiple techniques. This information will contribute to our understanding of plume dynamics and has relevance for local forecasts and global models.

General and specific comments:

1. This paper would benefit from reorganization with an aim toward concise communication of the study objectives, methods, results, and interpretation. Study objectives are stated a few times throughout the paper with slightly different levels of detail and emphasis (e.g. p.,3 L19-25, p.2 L31-33, P. 17 L17-19). A careful content and english

language edit would help cut down on redundancy, and tighten up the narrative. Attention to consistent use of language, terms, and nomenclatures through the different sections would help the readability. Decide on one spelling for sulfur versus Sulphur, for a single date format, etc. The paper is interesting and exciting, but is hard to digest in its current format. The introduction could be condensed, as there is extraneous information.

2. Gas section (section 5) and references to gas measurements. The plots and interpretation in this section could use some revision and clarification.

a. In plot 6, I don't see the pulse of SO2 observed at the end of phase 1 (noted in the section text and conclusions). Since Novac data can have strong anomalies due to atmospheric effects, wind, etc., it would be good to corroborate the novac data with emission rates from the mobile DOAS from Sept. 7, 11, 18 to confirm your observation. Plotting all the mobile data on figure 6 seems important.

b. It seems that the data in fig. 6 plot would be much easier to see if it were a scatter plot rather than column plot. E.g. in conclusions "During most of the eruption, SO2 fluxes have been lower than 1.5-2 kt day$-1$." It is actually hard to see that the red columns are in that range because of the error bars, which focus your eye on the max error bar value rather than the data points. Or is there some other reason you have it as a column plot?

c. Figure 6. caption: 'The uncertainty comes from the spectroscopic retrieval, radiative transfer, wind direction and speed, and plume height. This uncertainty is used in the computation of the daily mean values as presented in Figure 5.' Can you explain how this was done? Both the calculation of the uncertainty, and how it is used to calculate the daily mean values? Or send readers to a reference, if it is published elsewhere?

d. Fig. 7 –

i. The labeling/notations on the 2 FLIR images are inconsistent with each other, and

would be better if they were similar (e.g. you might have a single box for the max pixels in the image like for the bottom left image)

ii. Can you say something about the FLIR images, rather than just present them? Are they included to emphasize the less vigorous eruption during May as compared to august? Or is there another point you are wanting them to demonstrate?

iii. The Photo beneath the multi-gas plots detracts from the data plot, and should either stand on its own if you feel it is showing something of importance, or remove it. The plot axis labels cannot be read easily on the multigas plots, and need to be increased in size, and the plots presented in a larger format. Can you explain the trend in the different species, and if you think the concentrations make sense based on the plume traverse? e.g. Should the $SO_2$ and $CO_2$ anomalies be better correlated if they are from the plume, or are the instrument response times contributing to the lack of coincidence of peaks? Might you plot the C/S and $H_2O/CO_2$ that are described in the text? It is hard to take away anything from these plots in the current presentation.

iv. Are there some interesting differences in the multi-gas data for the 2 different eruption regimes (may versus august-oct)? might you show the data more clearly and completely since the text emphasizes this gas data?

v. Important to add emission rate for the $SO_2$ column amount profile plot. While this profile is interesting for people familiar with the technique, a plot of the mobile doas emission rates for the long eruption seems important in addition to this column amount plot.

3. Conclusions

a. The discussion of the preliminary data, and the relationship of the various data sets to each other, deserves its own section.

b. The emission rates for $CO_2$ and $H_2O$ are not reported in the paper, although it is referred to in the conclusion. It seems a table with the reported values scattered throughout the paper, and repeated in the conclusions could help the reader (gas emission rate data, Lidar coefficients, LR, particle numbers, etc.). I think such a table could be useful for others looking into plume dispersion and chemistry at their own volcanoes.

4. References – since you refer to radiative transfer a couple of times in the paper, it would be good to add a reference. Kern, C. et. al, 2012 (or other).

Minor comments:

1. It would be helpful for the maps to have a N arrow and a scale

2. Since you are reporting SO2 to 1 ppb, you might want to state the sensitivity and resolution of the pulse fluorescence SO2 analyzer.

3. p. 14 L 16-18. Your use of the terms 'course' and 'fine' to describe your particle size cut is unconventional for most of us who think of fine particles as PM2.5. Could you quality your description with a caveat like 'course particles as defined in this study'? Or use some other term to refer to the two size fractions you are discussing?

4. P. 14 L22. Figure 10 suggests SO2 is west of the vent, so the text is confusing since it states 'east'.

5. P. 14, L33. Do you mean 'volcanic aerosol-free air masses'? Otherwise, it is confusing – since particle size distribution in aerosol-free air masses doesn't make sense.

6. The red text on figure 1 is not legible. Can you use a color that more strongly contrasts, and with better resolution?

7. P. 16 L 7-8. Can you reorganize this sentence so that it is clearer? You could start the sentence with 'Examples of the evolution. . ..' And omit the first 5 words.

8. Figure 2. Might you Label the contour lines with elevation, for people not familiar with the topography? Fig.1 helps, but you could help your reader out by labeling it in fig. 2.

9. P. 16 L21-22. The wording of this sentence is unclear as you seem to be calling the sulphuric acid the precursor gas.

10. Can you mention the double maxima modeled in fig 9 bottom left in the final sentence of section 6? Or is it explained somewhere else? What might cause that?

11. For plots, state in captions or axis label if altitude is agl or asl

12. Is the 6.8 kt/d SO2 data point noted in the conclusion (and in the earlier text) on the plot?

Technical comments:

1. Identify acronyms with first use. While some sections do a good job of this, the Introduction needs attention. The subsequent sections don't have to repeat it, but watch for how the different authors use the acronyms so there is consistency throughout the paper. P. 15: ASQUA, ACTRIS – are these defined somewhere?

2. L26 p.3 –Do you mean topography rather than morphology?

3. L14-15 p. 4 – suggest revision of sentence: The Observatoire Volcanologique du Piton de la Fournaise (OVPF/IPGP) manages the monitoring networks on the island, allowing the observatory to follow eruptive and specific volcanic events, and to describe their time and space evolution.

4. L 17 p. 4- replace Internationals with International

5. P. 12-13, look carefully at the use of the word 'aerosols' versus 'aerosol' in this section.

6. P. 13 – both UTC and local time are provided in the discussion which is helpful. Consider doing this in key sections where you are describing a process that is dependent on diurnal orographic meteorology.

7. Global replace of 'pick up' with pick-up or 'pick-up truck'

8. Caption for fig. 10 – recommend clarifying sentence 2. "The flight path is coloured as a function of the measured. . ."

9. P.16 L26. This sentence needs to be clarified. '. . .because it depends whether the volcanic plume arrives at the station.' Do you mean it depends on 'when' it arrives? Or 'when and if' it arrives?

10. P. 16 L33-34. This sentence needs reorganization and grammar corrections.

11. P. 17 L2-4. This sentence needs to be rewritten, as it is very hard to follow.

12. Global replace 'researches' with 'research'

13. Figure 14. It would be kind to your readers to label the DMPS and AIS panels more clearly. Also, might want to make scale label and caption consistent (chose either cm-3 or #/cm)

14. Alternate wording suggestions have been included in a pdf version of the manuscript for many technical issues, but will not take the place of a thorough English language edit.

Please also note the supplement to this comment:

[revised manuscript text omitted]

---

## Author Comment (AC1) · 28 Feb 2017

1/ Main remarks

Comments from Referee

This paper strongly emphasized the gas composition and flux measurements but results presented somewhat fail to fulfill expectations: a) portable DOAS measurement is reported in the paper (L31 p4, L19 p7, L6 p9) but no corresponding result presented. Why? Figure 7 even present a transect across the plume. What is the corresponding $SO_2$ flux? Table 1 indicates 6 series of DOAS measurements but why no preliminary results presented ?

[Figure]

Author's response

As indicated in Table 1, the dataset of portable DOAS measurements is much smaller in comparison to the NOVAC dataset, which is discussed in detail in the text. We have added to Table 1 the SO2 fluxes calculated using the Salerno et al., 2009 approach. The (few) available portable data are in reasonable good agreement with NOVAC data, as it can be seen in Figure RC1 (attached).

Comments from Referee

b) August 2015 eruption is described as following 3 different phases, based on results from DOAS stations (L2-6 p10). But the uncertainties associated to these results are significant (Fig.6). - The eruption phase 1 described as associated to a progressive SO2 flux decreasing trend from 24/08 to 12/09 (L4 p10) is not convincing – this tendency is not clearly decreasing (Fig.6). These gas flux results (Fig.5 and Fig.5) will gain more strength if portable DOAS results are associated. - An "accelerating increase of SO2 flux between 13/09 and 18/10" is somewhat exaggerating. According to Fig.6 and accelerating tendency rather commenced in early October. Figure 6 indicates at least two strong degassing phases: from end august to mid-september and from early October to mid-October. Vigorous intermittent SO2 discharges were recorded between and after these two strong degassing phases.

Author's response

The referee points out correctly that Fig. 6 shows large uncertainty ranges and that, based on this figure, the interpretation of changes of activity seems questionable. However, the best estimate of daily SO2 emission, on which the interpretation of eruptive activity is based, is that of Fig. 5. The reason why they seem to differ, is that Fig. 6 shows the results of individual scan measurements that detected the plume. A careful uncertainty analysis was performed for each individual scan measurement, because of highly changing measurement conditions. For example, a plume can be observed completely above the horizon in one scan, and then decrease in altitude some minutes

later affecting the accuracy of the flux measurement. Presenting all scans with their un-certainty makes look the plot dominated by those measurements with large uncertainty (notice that a single day may have up to ïA¿50 scans), but we think the plot actually shows that the uncertainty varies among measurements and that some may indeed be quite large, presenting a challenge for interpretation. To compute a reasonable esti-mate of the daily mean value and its standard error (shown in Fig. 5), all valid scan measurements within a day are combined, weighting them according to their individual uncertainties, as explained succinctly in the caption of Fig. 6. By these approach, not only the mean value is more representative of the daily emission, but also the stan-dard error accounts for the fact that the larger the number of validated measurements, the more representative the statistic. In any case, we have remake Fig. 6 changing scale and using scatter points instead of columns, for better readability. Time evolu-tion of SO2 fluxes and their correlation with changes in magma bulk composition, lava flux, and other geophysical parameters during the August 2015 eruption has been dis-cussed in detail in Coppola et al., 2017 (EPSL); eruptive phases have been defined using this multidisciplinary approach; their description has been partly modified in this manuscript.

Comments from Referee

c) MultiGAS measurements is outline several time in the paper (L32 p8, L33 p10, L1 p11, L6 p11, L13 p11, L31 p17,. . .) and table indicates a total of around 8h of recording from May to October 2015. But curiously only 2 ratios are provided : H2O/CO2 = 50-240 (L12 p11) and CO2/SO2 <0.6 (L13 p 11).

Author's response

Reported values correspond to the measured ranges, not to two values. Text has been modified accordingly, and mean compositional data for the 2 distinct eruptive phases are now provided.

Comments from Referee

- It is well known that H2O and even CO2 are not easily measured in the plume. What is the error of this ratios ? A figure of the plots should be very informative. - Figure 7 gives concentration results which are not exploitable. The behaviour of H2O, CO2 and SO2 are totally different which may suggest no common source, that is surprising given that some of the measurement are performed close to the vent.

Author's response

Fig. 7 has been modified; in the original version, we wanted to emphasize the measurements of concentrations in situ by helicopter flight, which are informative for the meteorological community. We have now shown a typical ground based measurement performed closed to the vent, showing the occurrence of both correlated and uncorrelated peaks. Correlated peaks are indicative of a common source (volcanic degassing), while H2O-CO2 peaks (with no corresponding SO2 peak) imply contributions from ambient air (H2O) and/or low-t degassing features (CO2). The error in derived H2O/CO2 ratios is <20% in dense plume conditions, while it can increase up to 50% in dilute plumes, where the volcanic signal becomes limited compared to ambient air levels.

Comments from Referee

- Should we understand that H2O/CO2 and CO2/SO2 ratios are unchanged over the eruptive period ? That would be very surprising given the dynamic of the eruptive activity. Authors should add more results of multiGAS measurements and check the ratio changes which might describe better the eruption dynamic than the SO2 flux from the stationary DOAS.

Author's response

As a general comment, we want to stress that the detailed volcanological interpretation of the full dataset is not the main target of this paper; several other papers (e.g. Coppola et al., 2017) are under preparation, and they permit a complete analysis of each part of the dataset. In this paper, multiGAS data are only presented to give a

general idea of the plume composition near the vent and of the bulk fluxes; these parameters are fundamental to model plume ascent and dispersion; detailed discussion of the multiGAS dataset is the topic of a distinct paper (currently under preparation) which integrates a broader geochemical dataset.

Comments from Referee

- L32 p 4 indicates H2S was also measured. But curiously no result mentioned this gas. Is this suggest no H2S in the system ? That would be very surprising.

Author's response

H2S makes an irrelevant fraction of the S budget in the high-T vent emissions studied here, and was essentially below detection.

2/ Minor remarks

Comments from Referee - L25 p4: accumulation chamber for CO2 soil flux. Is this instrument deployed ? Not referring to in the rest of the paper. Add reference if developed elsewhere. Author's response As indicated in the text, CO2 fluxes are part of the measurements routinely performed by the OVPF observatory and they are part of the rich dataset acquired during each eruption of Piton de la Fournaise. Their presentation is not relevant here, as in this paper we focus on gas plume emission and dispersion.

Comments from Referee

- L21 p6, delete 2 after August.

Author's response

Thanks, it has been done.

Comments from Referee

- L21 p6, the date format is e.g., 2 August 2015 whilst L22 p6 the format is e.g., August 24, 2015. Harmonize date format throughout the paper.

Author's response

Thanks, it has been done.

Comments from Referee

L21 p6, to the south-southeast ?. . ..to the north ? L22 p6, to the west-southwest? What are these direction referred to ?

Author's response

The previous sentence specified that directions are referred to the Bory crater, we think it is obvious as it is presented.

Comments from Referee

- L31-32 the output budget ? Not calculated in the paper, why? Do add reference if done elsewhere.

Author's response

Budget have been computed. We added new sentences in the new version to discuss these results.

Author's changes in manuscript

In section 3.3Âǎ: ÂńÂǎWater is recalculated from hygrometric measurements. Subtraction of the atmospheric background permits the quantification of the elemental molar ratios (e.g. H2O/SO2, CO2/SO2 molar ratios) in the volcanic emissions. Correlation of these ratios with the SO2 fluxes (4.8+/-1.1 kt in May and 33.8+/-7.4 kt in August; Coppola et al., (2017)) measured by DOAS permit here a first estimation of the syn-eruptive fluxes of H2O and CO2 released by the eruptive vent(s).ÂǎÂż In section 4.2 ÂńÂǎThe combination of DOAS and MultiGAS permits to estimate that the May eruption emitted a minimum of 258 kt H2O 4.8 kt SO2 and 0.8 kt CO2, while the August-October eruption erupted 2649 kt H2O, 33.8 kt SO2 and 9.3 kt CO2.Âż

Comments from Referee - L12 p9, DOAS sessions are acquired with a high rate – what does it mean by high rate ?

Author's response

Thank you, we have specified the sampling rate of typically 5-10 min.

Comments from Referee

- L25-27 p9, 1870 t/d and 1840 t/d is that same if taking into account the errors. Thus not so sure that highest SO2 emission rate was observed on 20 May – maybe tone down this comparison.

Author's response

On the basis of our analysis, which treats carefully the uncertainty, we found that the mean flux measured on 20 May 2015 was indeed the highest in the record. Fig. 5 shows the respective standard errors for the interested reader.

Comments from Referee

- L31 p9, May SO2 fluxes are not in fig.6, but fig.5 – do modify the sentence.

Author's response

Thank you, the sentence was modified.

Comments from Referee

- L35 p9, add reference to the estimated 24-37 m3/s, or give further details if calculated in this work.

Author's response

The calculation has been performed in this work and the appropriate reference has been included. Reference been added.

Please also note the supplement to this comment:
http://www.atmos-chem-phys-discuss.net/acp-2016-865/acp-2016-865-AC1-
supplement.pdf

―――――――――――――――――

[Figure]

[Figure]

**Fig. 1.** Comparison of traverse mini-DOAS ('Portable DOAS') measurements with stationary scanning-DOAS ('NOVAC') measurements obtained on August 15 2015, evaluated with the methods mentioned in the manuscript.

---

## Author Comment (AC2) · 28 Feb 2017

Main comments

Comments from Referee

My main comment about this study is that I found it hard to follow - the introduction to the paper is verbose and discusses the motivation for the whole STRAP project rather than that specifically relevant to the results presented here. The results are separated out into separate sections that also describe methodological considerations for each method. Synthesis and comparison of the different results is not introduced until the conclusions, and is then very brief.

On page 17, line 16 (conclusions) the authors write that "The purpose of this article was twofold: (i) to present the methodological approach developed to track plume evolution from source to the distal area, and (ii) to summarize the preliminary observations of gaseous emissions, plume location, height and dispersion and gas-particle conversion" This would be a much clearer structure than the outline followed from page 3 lines 19-32 and currently followed. My suggestion is that the article is restructured on the following basis: 1) The introduction could be shortened and made more relevant to the results presented here, e.g., on page 3 lines 9 -19 are dominated by affiliations of the co-authors and some other information that could be in the acknowledgements; section 3.1 is long and could be condensed into an introduction to the more relevant material in section 2) The methods of the STRAP experiment, for which results are presented in this article, should be outlined in a methods sections. This should include a clear account of the temporal coverage of the observations that are being presented here (e.g., in a table or figure) It would help the article's readability if methods were clearly linked to the stated goal of tracking plume evolution from source to distal area. 3) The results should be described in a separate section and could be subdivided into (1) a presentation of preliminary observations of plume properties with clear reference to figures and (2) a synthesis of measurements relevant to understanding plume evolution. At present some of the results are well represented by figures, while others are described but not shown. 4) Discussion and conclusions should place the new observations made from measurements presented in this paper into context of past studies at Piton de la Fournaise and other volcanoes.

Author's response and general changes in manuscript

We agree that the paper will benefit with a re-organisation by separating the methods, results and discussions. It has been done taking into account the recommendations of both reviewers 2 and 3. Now the paper is constructed as follow:

1 - Introduction The introduction has been shortened.

[Figure]

2 - Description of the 2015 STRAP campaign on Piton de la Fournaise: We though that it is important to give in this section information about Reunion Island (meteorology conditions and topography), the geological context of Piton de la Fournaise, and to summarized the 4 eruptions of the STRAP campaign. The section 3.1 of the previous version has been condensed.

3 – Methods, models and measurements We have introduced a subsection named "Campaign management" to summarize the section 2.2 and to point out the location of the main sites of observations. We agree that most of the affiliations of 2.2 are not necessary in the text; they have been deleted and put in the acknowledgements. A subsection "Flexpart modelling" corresponding to section 4.1. A subsection named "Measurements near the plume source": this part integrates the description of the methods and instrumentation, previously introduced in section 5. A subsection "Measurements of the physical and chemical properties of the plume" which contains the technical elements and measurement methods introduced in the previous sections 6, 7 and 8.

4 - Preliminary results The results have been separated into three subsections of results and figures descriptions. "Simulation of the regional distribution in 2015": this part corresponds to section 4.2 "Plume geometry and gas emissions at the volcanic vent": this part corresponds to section 5 excluding the technical elements introduced in the new section 3. "Examples of volcanic plume distribution and chemical properties": this part groups the results of distal plume measurements at (sections 6, 7 and 8 of the previous version).

5- Discussion This new section has been purposed by both reviewer 2 and 3. This section contains the discussion of results previously introduced in the conclusion.

6 – Conclusion The conclusion has been modified and place the new observations made from measurements presented in this paper into the context of past studies at Piton de la Fournaise and other volcanoes.

Line by line comments:

Comments from Referee

Abstract: line 5 – do measurements span 85 days in total? Does this include gaps in activity?

Author's response

The STRAP campaign was conducted during all of the year 2015. 85 days represents the number of days of eruptive activity of the volcano and thus corresponds to the number of days of plume observations. We added Âń in the whole 2015 Âż in line 2 to be clearer and emphasize that the STRAP campaign occurred during the entire year of 2015.

Author's changes in manuscript

"The STRAP (Synergie Transdisciplinaire pour Répondre aux Aléas liés aux Panaches volcaniques) campaign was conducted in 2015..." by "The STRAP (Synergie Trans-disciplinaire pour Répondre aux Aléas liés aux Panaches volcaniques) campaign was conducted during the entire year of 2015..."

Comments from Referee

Abstract Line 11 – 'a particular emphasis is placed on...' this is an ambiguous phrase. Do you mean that this is a particularly interesting result?

Author's response

Yes, we wanted to emphasize this result. The sentence has been simplified to be less ambiguous. Author's changes in manuscript "A particular emphasis is placed on the gas-particle conversion with several cases of strong nucleation of sulphuric acid observed within the plume and at the distal site of the Maïdo observatory." by "Several cases of strong nucleation of sulphuric acid have been observed within the plume and at the distal site of the Maïdo observatory."

Comments from Referee

Abstract: How do the SO2, CO2 & H2O levels compare to past measurements/periods of activity. What are the implications for plume interaction with the atmosphere? Are there implications for understanding the development of the eruption (e.g., from increase in SO2 at end of phase referred to later?)

Author's response

The PdF emission are negligible outside eruptive period (see review of Di Muro et al., 2016). So the concentration level of volcanic pollutant (gas and aerosols) are several times lower than during the eruptions. Author's changes in manuscript We added "During the last decades, the degassing of Piton de la Fournaise was negligible outside the eruption periods." in section "Geological context of Piton de la Fournaise".

Comments from Referee

I understand from Section 3.2 that observations from 20th June 2014 to October 2015 are presented in the paper, but from the Figures (Especially 12 and 13) it looks like data were only acquired in 2015 ( the abstract refers to 85 days of measurements and from page 3 line 28 (and figures) it sounds like only the climatology for two eruptions is described). Overall I found it difficult to get my head around the differences in temporal coverage of all the different measurement types – I suggest that the authors include a table, or perhaps a figure, to compare the duration and temporal coverage of each type of observation.

Author's response

You are right, the section 3.2 is confusing by including a part of 2014 in the STRAP period. It has been corrected in the new version. As explain above, the STRAP campaign only occurred during the year 2015. However, the OVPF managed continuously all eruptions of the Piton de la Fournaise.

We did not find the way to summarize all the observations on one figure due to the

disparity of measurements types and their duration. We have chosen to introduce four tables (two in the main text table 1 and table 2 and two in the appendix table A1 and table A2), and one figure for the Maido observatory (permanent observation).

Comments from Referee

Page 2 line 7-8. Use of 'on one side', 'on the other side' is confusing – these are not opposing ideas? Author's response Thanks, it has been corrected.

Author's changes in manuscript

The new sentences are: " Improving our ability to quantify and model the genesis, dispersion and impact of a volcanic plume is thus a key challenge for scientists and societal stakeholders. Furthermore, mitigation of volcanic crisis relies on efficient, and effective, communication and interaction between multidisciplinary scientific actors in geology, physics, chemistry, and remote sensing."

Comments from Referee

Page 2 line 18. Please add reference for impossibility of obtaining source parameters at Eyja? Author's response It has been done: the reference of Ripepe et al., 2013 is added.

Comments from Referee

Page 2 line 31: 'an objective' - is this the particular aim of this work? The following sentence refers to real-time measurement, which I think is not necessary for these goals.

Author's response

This is true, "Real-Time" has been deleted.

Comments from Referee

page 3, line 5: I don't think this is true. The Boulon 2011 paper does not include mea-

surements made within a volcanic plume. And there are certainly other earlier studies that present measurements of aerosol within volcanic plumes (e.g., Mather et al., 2004; Rose et al., 2006; Martin et al., 2008). Although it provides no information about nucleation mechanism Ebmeier et al., 2014 also shows that there is elevated aerosol and depressed cloud droplet size downwind of PdlF in satellite retrievals averaged over a decade (and a greater effect for periods of eruption). It would be interesting to know how these course observations compare to your much more detailed multi-sensor measurements.

Author's response

The observation of the reviewer is valid. There is an error in the reference of Boulon et al., 2011. Boulon Julien, Karine Sellegri, Maxime Hervo and Paolo Laj, Observations of nucleation of new particles in a volcanic plumeÂăÂż, PNAS, July 11, doi: 10.1073/pnas.1104923108, 2011 To our knowledge, the paper is the first where measurement of ultra-fine particles (sub 5 nm, which characterize the gas-particles nucleation process) have been made within a volcanic plume. However these measurements have been made far from the volcanic vent (in France on a plume from Island). Here we probably present the first observation of ultra-fine particles observed at few km of the vent (AIS instrument). The concentration are thus much higher in our study thus adding a new perspective to the work of Boulon et al., 2011. The reference has been corrected in the new version of the article.

We also added the reference of Ebmeier et al., 2014 in the introduction (thanks for this interesting paper). There were no cloud droplet measurement during the STRAP campaign. So it is not possible to compare our results with the study of Ebmeier et al. at this stage. However, cloud resolving models (MesoNH, see Durand et al., 2014, jgr) will be applied in some of the case studies of the STRAP period. One focus will be on evaluating the aerosol activation (small cloud droplet formation) downwind of the Piton de la Fournaise vent. We hope to simulate the same process of cloud formation associated with a volcanic plume composed by high number of CCN. This new study
could then be compared to satellites observations of Ebmeier et al., 2014.

The references to the papers of Mather et al., 2005; Rose et al., 2006, and Martin et al., 2008 have been added to Robock, 2000.

Comments from Referee

Page 4, line 2: 'unique and craggy' is uninformative, 'benefits from a tropical climate softened by the breezes of the Indian Ocean' is also rather informal in style.

Author's response

You are right about the rather vague meaning of the terms "unique" and "softened". The term craggy was used to emphasize that the topography of the Island is steep and the local circulation is complex in the valley. The sentences have been modified.

Comments from Referee

Section 2.2 title: 'means'=methods?

Author's response

We wanted to refer to capabilities in instrumentation. Nevertheless, the word "means" and the title have been removed with the reorganisation of the paper, described above.

Comments from Referee

Page 4 line 19: what kind of imagery? Photographs? Figure 1: Resolution appears to be quite low for the size of Figure.

Author's response

Visible and IR imagery are routinely acquired. The text has been modified accordingly. Figure 1 has been modified: the colours of characters were changed, the resolution was increased.

Comments from Referee

Caption: Page 5 line 20 range of dates is surprisingly precise

Author's response

We don't understand this comment. There is no range of dates specified in page 5 line 20.

Comments from Referee

page 10 line 10: I'm not sure that this is an acceleration?

Author's response

The word acceleration is probably not appropriate. In the new version, the word "accelerate" has been changed into "increase".

Comments from Referee

Page 17 line 21 – where are these geometries shown?

Author's response

Fig. 4 summarizes the results of plume height and direction measurements by the NOVAC instruments and ground Meteo-France stations.

Comments from Referee

Page 18 line 10 → rugged?

Author's response

Thanks, it has been corrected.

Comments from Referee

Conclusions: Comparison to the previous level of knowledge about the PdlF plume would be useful here – both to place your results in context and help the reader appreciate the level of advance in knowledge offered by such an integrated multi-

[Figure]
**Interactive comment**

methodological approach.

Author's response

Piton de la Fournaise is one of the most active volcanoes in the World. In spite of that, very little is known about its gas emissions in terms of fluxes, composition and evolution in time and space (see the recent review of Di Muro et al., 2016). The 2015 experiment provides the first complete complete characterisation of gas emissions of Piton de la Fournaise. Interestingly, the experiment captured two distinct end-members of the typical PdF activity: i) a fast and exponentially declining eruptive activity (May 2015) and ii) a complex, long lasting and large volume eruption (August-October eruption). These elements have been added in the new section "Discussion".

Comments from Referee

Through the article there are English phrases that are ambiguous and some rather awkward constructions. I suggest that English language proof reading would help the final version of this article.

Author's response

We have done our best to improve the English. The new version of the paper has been read by a native English speaker. Your review and help was strongly appreciated.

Please also note the supplement to this comment:
http://www.atmos-chem-phys-discuss.net/acp-2016-865/acp-2016-865-AC2-supplement.pdf

---

## Author Comment (AC3) · 28 Feb 2017

[revised manuscript text omitted]

---

## Author Comment (AC4) · 28 Feb 2017

The new version of the paper is in the supplement file.

Please also note the supplement to this comment:
http://www.atmos-chem-phys-discuss.net/acp-2016-865/acp-2016-865-AC4-supplement.pdf

---

## Author Comment (AC5) · 28 Feb 2017

General and specific comments:

Comments from Referee1.

This paper would benefit from reorganization with an aim toward concise communication of the study objectives, methods, results, and interpretation. Study objectives are stated a few times throughout the paper with slightly different levels of detail and emphasis (e.g. p., 3 L19-25, p.2 L31-33, P. 17 L17-19). A careful content and english language edit would help cut down on redundancy, and tighten up the narrative. Attention to consistent use of language, terms, and nomenclatures through the different

sections would help the readability. Decide on one spelling for sulfur versus Sulphur, for a single date format, etc. The paper is interesting and exciting, but is hard to digest in its current format. The introduction could be condensed, as there is extraneous information.

Author's response

We have done our best to improve the English and your help was strongly appreciated (thanks for the supplement comments). We have tried to clarified the general objectives of more interest to the atmospheric community (p2 L31-32), the purpose of the paper (p., 3 L19-25), and deleted the repetition of the paper's objective in the conclusion(P. 17 L17-19).

We have taken more attention to present a more consistent spelling and format. The new version of the paper has also been read by a native English speaker.

We agree that the paper will benefit with a re-organisation by separating the methods, results and discussions. It has been done taking into account the recommendations of both reviewers 2 and 3. Now the paper is constructed as follow: 1 – IntroductionÂă: The introduction has been shortened. 2 - Description of the 2015 STRAP campaign on Piton de la Fournaise: We though that it is important to give in this section information about Reunion Island (meteorology conditions and topography), the Piton de la Fournaise volcano characteristics, and to summarized the 4 eruptions of the STRAP campaign. The section 3.1 of the previous version has been condensed. 3 – Methods, models and measurements We have introduced a subsection named "Campaign management" to summarize the section 2.2 and to point out the location of the main sites of observations. We agree that most of the affiliations of 2.2 are not necessary in the text; they have been deleted and put in the acknowledgements. A subsection "Flexpart modelling" corresponding to section 4.1. A subsection named "Measurements near the plume source": this part integrates the description of the methods and instrumentation, previously introduced in section 5. A subsection "Measurements of the physical and

chemical properties of the plume" which contains the technical elements and measurement methods introduced in the previous sections 6, 7 and 8. 4 - Preliminary results The results have been separated into three subsections of results and figures descriptions. "Simulation of the regional distribution in 2015": this part corresponds to section 4.2 "Plume geometry and gas emissions at the volcanic vent": this part corresponds to section 5 excluding the technical elements introduced in the new section 3. "Examples of volcanic plume distribution and chemical properties": this part groups the results of distal plume measurements at (sections 6, 7 and 8 of the previous version). 5- Discussion This new section has been purposed by both reviewer 2 and 3. This section contains the discussion of results previously introduced in the conclusion. 6 – Conclusion The conclusion has been modified and place the new observations made from measurements presented in this paper into the context of past studies at Piton de la Fournaise and other volcanoes.

Comments from Referee

2. Gas section (section 5) and references to gas measurements. The plots and interpretation in this section could use some revision and clarification a. In plot 6, I don't see the pulse of SO2 observed at the end of phase 1 (noted in the section text and conclusions). Since Novac data can have strong anomalies due to atmospheric effects, wind, etc., it would be good to corroborate the novac data with emission rates from the mobile DOAS from Sept. 7, 11, 18 to confirm your observation. Plotting all the mobile data on figure 6 seems important. b. It seems that the data in fig. 6 plot would be much easier to see if it were a scatter plot rather than column plot. E.g. in conclusions "During most of the eruption, SO2 fluxes have been lower than 1.5-2 kt day$-1$." It is actually hard to see that the red columns are in that range because of the error bars, which focus your eye on the max error bar value rather than the data points. Or is there some other reason you have it as a column plot? c. Figure 6. caption: 'The uncertainty comes from the spectroscopic retrieval, radiative transfer, wind direction and speed, and plume height. This uncertainty is used in the computation of the daily mean values

as presented in Figure 5.' Can you explain how this was done? Both the calculation of the uncertainty, and how it is used to calculate the daily mean values? Or send readers to a reference, if it is published elsewhere?

Author's response

a. $SO_2$ fluxes obtained by portable DOAS and calculated using the Salerno et al., (2009) approach have been added to Table 1. Even if it is tricky to compare average daily fluxes (NOVAC; fig. 5) and multiple daily scans (NOVAC; fig. 6) with single portable traverses, both methods are in reasonable good agreement (Table 1), as it can be seen in Figure (below):

a, b. The referee points out correctly that Fig. 6 shows large uncertainty ranges and that, based on this figure, the interpretation of changes of activity seems questionable. However, the best estimate of daily $SO_2$ emission, on which the interpretation of eruptive activity is based, is that of Fig. 5. The reason why they seem to differ, is that Fig. 6 shows the results of individual scan measurements that detected the plume. A careful uncertainty analysis was performed for each individual scan measurement, because of highly changing measurement conditions. For example, a plume can be observed completely above the horizon in one scan, and then decrease in altitude some minutes later affecting the accuracy of the flux measurement. Presenting all scans with their uncertainty makes look the plot dominated by those measurements with large uncertainty (notice that a single day may have up to ïA¿50 scans), but we think the plot actually shows that the uncertainty varies among measurements and that some may indeed be quite large, presenting a challenge for interpretation. To compute a reasonable estimate of the daily mean value and its standard error (shown in Fig. 5), all valid scan measurements within a day are combined, weighting them according to their individual uncertainties, as explained succinctly in the caption of Fig. 6. By these approach, not only the mean value is more representative of the daily emission, but also the standard error accounts for the fact that the larger the number of validated measurements, the more representative the statistic. In any case, we have remake Fig. 6 changing scale

and using scatter points instead of columns, for better readability.

c. There is an unpublished PhD thesis (Arellano et al., 2014, Chalmers University of Technology) describing in detail the methodology behind the uncertainty calculation and the computation of daily statistics. Each scan has its own uncertainty analysis based on sampling of distributions of error for each of the variables (column density, wind speed, plume height, plume direction). The daily average is calculated as a weighted mean that favours measurements with lower uncertainty. The standard error is calculated taken into account the individual uncertainties and of course the number of valid measurements on each day. We did not abound in details in the manuscript to avoid giving too much emphasis to this technique, but included Fig. 5 to show that obtaining daily statistics is not a simple matter of averaging measurements because the quality of the measurements may vary considerably even within a single day.

Comments from Referee

i. The labeling/notations on the 2 FLIR images are inconsistent with each other, and would be better if they were similar (e.g. you might have a single box for the max pixels in the image like for the bottom left image)

ii. Can you say something about the FLIR images, rather than just present them? Are they included to emphasize the less vigorous eruption during May as compared to august? Or is there another point you are wanting them to demonstrate?

iii. The Photo beneath the multi-gas plots detracts from the data plot, and should either stand on its own if you feel it is showing something of importance, or remove it. The plot axis labels cannot be read easily on the MultiGAS plots, and need to be increased in size, and the plots presented in a larger format. Can you explain the trend in the different species, and if you think the concentrations make sense based on the plume traverse? e.g. Should the $SO_2$ and $CO_2$ anomalies be better correlated if they are from the plume, or are the instrument response times contributing to the lack of coincidence of peaks? Might you plot the C/S and $H_2O/CO_2$ that are described in the text? It is

hard to take away anything from these plots in the current presentation.

iv. Are there some interesting differences in the multi-gas data for the 2 different eruption regimes (May versus August-October)? might you show the data more clearly and completely since the text emphasizes this gas data?

v. Important to add emission rate for the SO2 column amount profile plot. While this profile is interesting for people familiar with the technique, a plot of the mobile doas emission rates for the long eruption seems important in addition to this column amount plot.

Author's response

Fig. 7 has been modified.

i. Two new IR images of the beginning of the August 2015 eruption have now been included with the aim at highlighting the fast evolution from linear to spot source for the plume emission. The text has been modified accordingly.

ii. The MultiGAS figure has been changed; the original image of the results obtained by helicopter flight has been replaced with a typical ground based measurement performed in near field close to the high temperature source; correlated peaks in MultiGAS measurements have been evidenced.

iii. MultiGAS data show relatively moderate change in time, as discussed in the text; however their detailed presentation and interpretation is the topic of a distinct paper (in preparation) which integrates a larger geochemical dataset (bulk rocks; melt inclusion; mineral phase equilibria; gas fluxes and molar ratios)

iv. SO2 emission rates estimated using mobile DOAS have been reported in Table A2.

v. Emission rate has been on the SO2 column amount profile plot (Fig. 7).

Comments from Referee

3. Conclusions a. The discussion of the preliminary data, and the relationship of the various data sets to each other, deserves its own section.

b. The emission rates for CO2 and H2O are not reported in the paper, although it is referred to in the conclusion. It seems a table with the reported values scattered throughut the paper, and repeated in the conclusions could help the reader (gas emission rate data, Lidar coefficients, LR, particle numbers, etc.). I think such a table could be useful for others looking into plume dispersion and chemistry at their own volcanoes.

4. References – since you refer to radiative transfer a couple of times in the paper, it would be good to add a reference. Kern, C. et. al, 2012 (or other).

Author's response

The discussion of the preliminary data has been now put in a separate section (section 5). It is complicated to summarize all the observations in one table due to the disparity of measurements types and their duration. We have chosen to introduce four tables (two in the main text table 1 and table 2 and two in the appendix table A1 and table A2), and one figure for the Maido observatory (permanent observation). It is not possible to summarize in one table all LIDAR measurements (more than 600 profiles). A dedicated paper is in preparation. We hope that the re-structuration of the paper will be clearer for readers.

The reference to Kern et al., 2010 has been added in the corresponding mention to radiative transfer effects and the reference list.

Minor comments:

Comments from Referee

1. It would be helpful for the maps to have a N arrow and a scale

Author's response

The new Figure 1 have a scale in the bottom part of the picture and a N arrow.

Comments from Referee

2. Since you are reporting SO2 to 1 ppb, you might want to state the sensitivity and resolution of the pulse fluorescence SO2 analyzer.

Author's response

The limit of detection of the SO2 analyzer is 50 ppt (0.05 ppb). This instrument is used for air quality studies and it is able to measure low level concentrations (e.g. at the free troposphere or rural areas). The sentence has been modified

Author's changes in manuscript

The new sentence is: "Gas phase measurements of sulphur dioxide were made using a UV Fluorescence SO2 Analyzer (Teledyne, model T100U), which relies on pulsed fluorescence and has a detection limit of 50 ppt (i.e. 0.05 ppb)."

Comments from Referee

3. p. 14 L 16-18. Your use of the terms 'course' and 'fine' to describe your particle size cut is unconventional for most of us who think of fine particles as PM2.5. Could you quality your description with a caveat like 'course particles as defined in this study'? Or use some other term to refer to the two size fractions you are discussing?

Author's response

In atmospheric aerosol science fine particles are related to sub-micron-size particles (PM1 with diameter $< 1\mu$m). Ultra-fine particles characterize aerosols with a size below the accumulation mode (diameter $< 100$ nm). The coarse particles have diameter greater than 1 $\mu$m.

The AIS and the nanoCPC system installed at the Maïdo observatory are able to count particle number at 5 nm (nucleation mode). We agree that we are not in the range of particles usually observed in volcanic sources (e.g. supermicronic size - coarse mode for ashes). This is the reason why we have given precise size information in the paper,

as follows:

- ultrafine particles (Dp < 100 nm) - fine particles (Dp < 1 $\mu$m) - coarse particle (Dp > 1 $\mu$m)

Author's changes in manuscript

New sentences are:

"Meanwhile, there was a moderate increase in coarse particle concentrations (particle diameter Dp > 1 $\mu$m)."

"It is very likely that the particles in the volcanic plume were generated by oxidation of volcanic SO2 and subsequent particle nucleation or by condensation of volatile compounds onto pre-existing fine particles (Dp < 1 $\mu$m)."

"The morning advection of a relatively wide range of ultrafine particles (Dp < 100 nm) to the Maïdo station indicates that nucleation and early growth takes place already at the vicinity of the crater, and continues within the plume at least up to the Ma\"ido station."

Comments from Referee

4. P. 14 L22. Figure 10 suggests SO2 is west of the vent, so the text is confusing since it states 'east'.

Author's response

It was an error (thanks). The sentence has been corrected in the new version.

Comments from Referee

5. P. 14, L33. Do you mean 'volcanic aerosol-free air masses'? Otherwise, it is confusing - since particle size distribution in aerosol-free air masses doesn't make sense.

Author's response

We mean "volcanic aerosol-free air masses". Thanks for this remark. Is has been

corrected in the new version.

Comments from Referee

6. The red text on figure 1 is not legible. Can you use a color that more strongly contrasts, and with better resolution?

Author's response

You are right. The colour of the text has been changed to white and the size of the figure increases.

Comments from Referee

7. P. 16 L 7-8. Can you reorganize this sentence so that it is clearer? You could start the sentence with 'Examples of the evolution. . ..' And omit the first 5 words.

Author's response

The sentence has been modified.

Author's changes in manuscript

Examples of the fast growth of cluster ions to larger sizes can be followed on the SMPS size distributions up to 50 nm on 1-2 September, and 100 nm on 20-21 May.

Comments from Referee

8. Figure 2. Might you Label the contour lines with elevation, for people not familiar with the topography? Fig.1 helps, but you could help your reader out by labeling it in fig. 2.

Author's response

It has been done for figures 2, 3, 8, 9 and 10. Now dotted lines represents in red the topography at 2000 m asl and in black the topography at 1000 m asl.

Comments from Referee

9. P. 16 L21-22. The wording of this sentence is unclear as you seem to be calling the sulphuric acid the precursor gas.

Author's response

The word precursor is deleted.

Author's changes in manuscript

The sentence is now: "Due to its low saturated vapour pressure under typical atmospheric temperatures (Marti et al., 1997), the common assumption in the scientific community is that the sulphuric acid is the main gas responsible for the nucleation processes."

Comments from Referee

10. Can you mention the double maxima modelled in fig 9 bottom left in the final sentence of section 6? Or is it explained somewhere else? What might cause that?

Author's response

This double maxima is related to the modification of wind intensity above the vent. So the volcanic air mass loads different quantity of volcanic pollutant during its passage above the emission aera.

Author's changes in manuscript

The new sentences are: "On 2 September 2015, the plume was forecasted to be located north-west of the volcano. Two maxima were modelled by FLEXPART (above the OVPF and above the Maïdo area) in relation with the evolution of the wind intensity above the vent."

Comments from Referee

11. For plots, state in captions or axis label if altitude is agl or asl

Author's response

It has been corrected.

Comments from Referee

12. Is the 6.8 kt/d SO2 data point noted in the conclusion (and in the earlier text) on the plot?

Author's response

Figure 6, which shows individual scan measurements shows these values, as discussed in the text. See above for changes done on this figure for better readability.

Technical comments:

Comments from Referee

1. Identify acronyms with first use. While some sections do a good job of this, the Introduction needs attention. The subsequent sections don't have to repeat it, but watch for how the different authors use the acronyms so there is consistency throughout the paper. P. 15: ASQUA, ACTRIS – are these defined somewhere?

Author's response

ASQUA, VACC are deleted. ACTRIS is now defined.

Author's changes in manuscript

"The quality of the DMPS measurements was checked for flow rates and relative humidity according to the ACTRIS (Aerosols, Clouds, and Trace gases Research InfraStructure Network) recommendations (Wiedensohler et al., 2012)."

Comments from Referee

2. L26 p.3 –Do you mean topography rather than morphology?

Author's response

Yes, it has been corrected.

Comments from Referee

3. L14-15 p. 4 – suggest revision of sentence: The Observatoire Volcanologique du Piton de la Fournaise (OVPF/IPGP) manages the monitoring networks on the island, allowing the observatory to follow eruptive and specific volcanic events, and to describe their time and space evolution.

Author's response

It has been modified.

Comments from Referee

4. L 17 p. 4- replace Internationals with International

Author's response

Thanks, this sentence has been deleted in the new version.

Comments from Referee

5. P. 12-13, look carefully at the use of the word 'aerosols' versus 'aerosol' in this section.

Author's response

We have corrected it in the text.

Comments from Referee

6. P. 13 – both UTC and local time are provided in the discussion which is helpful. Consider doing this in key sections where you are describing a process that is dependent on diurnal orographic meteorology.

Author's response

It has been done.

[Figure]

Comments from Referee

7. Global replace of 'pick up' with pick-up or 'pick-up truck'

Author's response

In the new version we have changed "pick up" by "pick-up truck".

Comments from Referee

8. Caption for fig. 10 – recommend clarifying sentence 2. "The flight path is coloured as a function of the measured..."

Author's response

Thanks, the sentence of the caption has been modified.

Comments from Referee

9. P.16 L26. This sentence needs to be clarified. '. . .because it depends whether the volcanic plume arrives at the station.' Do you mean it depends on 'when' it arrives? Or 'when and if' it arrives?

Author's response

We agree that the sentence was unclear. The new sentence is "Unlike other parameters, for instance anthropogenic pollutants, the SO2 concentration variation is not periodic because it depends on whether the volcanic plume is advected to the station or not"

Comments from Referee

10. P. 16 L33-34. This sentence needs reorganization and grammar corrections.

Author's response

The sentence has been rephrased as "For the case of 20 May, it is possible that newly formed particles are grown by condensation to sizes above the detection limit of our

instrumentation."

Comments from Referee

11. P. 17 L2-4. This sentence needs to be rewritten, as it is very hard to follow.

Author's response

The sentence is now written as: "Then the variability of the correlation between the new particle formation rate and sulphuric acid will be further studied for other case studies. This will allow to derive, for the first time to our knowledge, a parameterization of nucleation rate specific to volcanic plumes."

Comments from Referee

12. Global replace 'researches' with 'research'

Author's response

Thanks, it has been done.

Comments from Referee

13. Figure 14. It would be kind to your readers to label the DMPS and AIS panels more clearly. Also, might want to make scale label and caption consistent (chose either cm-3 or #/cm)

Author's response

The Figure 14 has been modified. We have chosen cm-3 to be consistent with the rest of the paper.

Comments from Referee

14. Alternate wording suggestions have been included in a pdf version of the manuscript for many technical issues, but will not take the place of a through English language edit.

Author's response

Thanks for this work. It was very helpful. We have done our best to improve the English and the new version of the paper has now been read by a native English speaker.

Comments from Referee

(Table A1): the numbering of this table seems unusual.

Author's response

This numbering came from latex Copernicus package (it is automatic). So it is the format asked by the ACP. Table 1, 2 are numbering for the main text whereas A1, A2 corresponds to appendix.

Comments from Referee

"One explanation could be attributed to a subsidence..." : grounding? or sinking? subsidence generally refer to solid surfaces

Author's response

In meteorology the term "subsidence" is largely used to refer to downward transport of air masses.

Comments from Referee

"One can also notice a fresh crossover of aerosols plume starting at 11 UTC, credibly coming directly from the vent."Ăă: introduction?

Author's response

We are not sure to understand this comment placed in the supplement document.

Author's changes in manuscript

We purpose to rephrase as: "The LIDAR backscattered signal increases from 2.2 up to 2.6-3.5 (au) between 0 and 500 m agl at 10:30 UTC (14:30 h LT) until the end of

the measurement period. This shows the passage of a freshly emitted aerosols plume likely coming from the vent."

Please also note the supplement to this comment:
http://www.atmos-chem-phys-discuss.net/acp-2016-865/acp-2016-865-AC5-supplement.pdf

[Figure]

[Figure]

**Fig. 1.** Comparison of traverse mini-DOAS ('Portable DOAS') measurements with stationary scanning-DOAS ('NOVAC') measurements obtained on August 15 2015, evaluated with the methods mentioned in the manuscript.

---

## Author Comment (AC6) · 28 Feb 2017

The new version of the paper is in the supplement file

Please also note the supplement to this comment:
http://www.atmos-chem-phys-discuss.net/acp-2016-865/acp-2016-865-AC6-supplement.pdf